# The impact of atmospheric blocking on the compounding effect of ozone pollution and temperature: A copula-based approach

Noelia Otero [1,2,3], Oscar E. Jurado [2], Tim Butler [1,2], and Henning W. Rust [2]

[1]Institute for Advanced Sustainability Studies, Potsdam, Germany
[2]Institut für Meteorologie, Freie Universität Berlin, Germany
[3]Now at Oeschger Centre for Climate Change Research (OCCR), Bern, Switzerland

**Correspondence:** Noelia Otero (noelia.otero@giub.unibe.ch)

**Abstract.** Ozone pollution and high temperatures have adverse health impacts that can be amplified by the combined effects of ozone and temperature. Moreover, changes in weather patterns are expected to alter ozone pollution episodes and temperature extremes. In particular, atmospheric blocking is a high-impact large-scale phenomenon at mid-high latitudes that has been associated with temperature extremes. This study examines the impact of atmospheric blocking on the ozone and temperature dependence among measurement stations over Europe during the period 1999-2015. We use a copula-based method to model the dependence between both variables under blocking and non blocking conditions. This approach allows us to examine the impact of blocks on the joint probability distribution. Our results showed that blocks lead to an increasing strength in the upper tail dependence of ozone and temperature extremes (>95th) in north-west and central Europe (e.g., the UK, Belgium, Netherlands, Luxembourg, Germany and north-west of France). The analysis of the probability hazard scenarios revealed that blocks generally enhance the probability of compound ozone and temperature events by 20% in a large number of stations over central Europe. The probability of ozone or temperature exceedances increases 30% (on average) under the presence of atmospheric blocking. Furthermore, we found that, in a number of stations over north-western Europe, atmospheric blocking increases the probability of ozone exceedances by 30% given high temperatures. Our results point out the strong influence of atmospheric blocking on the compounding effect of ozone and temperature events, suggesting that blocks might be considered as a relevant predicting factor when assessing the risks of ozone-heat related health effects.

## 1 Introduction

Air pollution and heat waves pose a serious risk to health globally (Analitis et al., 2014; WHO, 2015) and evidence suggests that when extreme weather and air pollution episodes occur in coincidence, their health effects are nonlinearly amplified beyond the sum of their individual effects (Willers et al., 2016). Climate change is expected to increase the probability of heat extremes (Seneviratne et al., 2014) and alter air quality (Doherty et al., 2018). Tropospheric ozone is recognised as a harmful pollutant with negative impacts not just on human health, but also on ecosystems (EEA, 2019). Tropospheric ozone is a secondary pollutant formed from complex photochemical reactions of nitrogen oxides ($NO_x$), carbon monoxide (CO) and volatile organic compounds (VOCs) (Seinfeld and Pandis, 2006). The combination of emissions of ozone precursors and specific weather conditions, such as high temperatures, low wind and persistent slow-moving high-pressure systems, favour

high ozone pollution episodes (Jacob et al., 1993). Temperature has been identified as one of the main meteorological drivers of high ozone episodes in polluted regions over the US (Porter et al., 2015) and most of central Europe (Otero et al., 2016).

Several studies examined the relationship between ozone and temperature extremes and their joint occurrences over the US (Shen et al., 2016; Phalitnonkiat et al., 2018). Phalitnonkiat et al. (2018) measured the joint extremal dependence of ozone and temperature using the spectral dependence of their extremes. They found that temperature and ozone were, overall, well correlated across many areas across the US, but noted a reduced correlation when examining the tail of the distribution. Schnell and Prather (2017) examined the co-occurrence of extreme temperatures and air pollution (ozone and fine particulate matter) and found temperature extremes to be consistently associated in space and time with high levels of ozone over the contiguous US. Sun et al. (2017) estimated a 50% of conditional probability of high ozone levels given the high temperatures in the north-eastern US, while less than 20% was found in the western US. Zhang et al. (2017) compared ozone levels during extreme and non-extreme weather events and reported higher ozone levels during extreme weather events, including heat waves, atmospheric stagnation and their compound extremes over the US. Specifically, they pointed out an enhancement of ozone concentrations when heat waves and atmospheric stagnation events occur simultaneously. Recently, Hertig et al. (2020) analysed combined episodes of heat and ozone pollution waves in two European regions (Germany and Portugal) and their association with mortality rates. This study confirmed the strong impact of compounded heat-ozone waves on excess mortality rates in those regions.

The co-occurrence of extremes are known as a combination of extreme events, which can potentially have a greater impact than the independent hazard event (Zscheischler and Seneviratne, 2017). The compounding effects from high temperature and ozone pollution levels greatly increase the risk to human health (Hertig et al., 2020). Furthermore, the extremes of temperature and high ozone episodes might be exacerbated by underlying climatological drivers (Schnell and Prather, 2017). Large-scale atmospheric circulation is a key driving factor of the variability of surface meteorological variables, including air temperature and extreme temperature events (Pfahl., 2014) and plays an important role in air quality (Russo et al., 2014; Hertig et al., 2020). Extreme weather events are closely linked to anomalies of the atmospheric circulation that can be categorised by 'weather regimes' such as cyclones and atmospheric blocking (Pfahl., 2014). For instance, the extreme temperatures and lack of precipitation during the summer of 2003 in Europe have been related to the persistent anticyclonic conditions over central Europe (Fink et al., 2004; Solberg et al., 2008). This particular episode led to exceptionally long-lasting and spatially extensive periods of high levels of ozone pollution over Europe (Fiala et al., 2003). Dole et al. (2011) suggested that the persistent blocking of westerly flow was essential during the 2010 heatwave in Russia that killed tens of thousands of people.

Atmospheric blocking is a large-scale phenomenon defined by persistent anticyclones that interrupt the westerly flow in mid-latitudes (Barriopedro et al., 2006), and has been associated with extreme temperature events (Sillmann et al., 2011; Brunner et al., 2017). Pfahl and Wernli (2012) showed that warm temperature extremes often co-occur with atmospheric blocking at the same location and, recently, Röthlisberger and Martius (2019) found that atmospheric blocking also increases the persistence of periods with hot and dry weather conditions that occur concomitantly during summer. A few studies have examined the impact of blocks on air pollution. Ordóñez et al. (2017) focused on the regional responses of maximum daily average of 8h ozone (MDA8O$_3$) to the persistence of blocks and ridges over Europe. They showed that blocks within the European sector (defined

as 0-30° E) led to positive anomalies of MDA8O$_3$ over central Europe in spring and summer and found that a considerable proportion of the variability of MDA8O$_3$ exceedances could be explained by blocking patterns. Cai et al. (2020) analysed the influence of persistent blocking conditions on several high pollution episodes of fine particulate matter over northern China. They showed that blocking structures lead to 62.5% of persistent air pollution events during winter in northern China and pointed out that blocks might be used as an indication of persistent heavy air pollution in that location during winter.

The significant linkage between warm extremes and blocking and the strong temperature dependence of ozone motivates the present work that aims to assess the impact of persistent blocks on the compounding effect of ozone and temperature over Europe. While previous studies have examined the relationship between extremes of surface ozone and temperature (Schnell and Prather, 2017) and have provided a comprehensive analysis of seasonal impacts of blocks on European surface ozone pollution (e.g Ordóñez et al., 2017), we present, for the first time (to the authors, knowledge), a quantification of the effect of blocks

on the co-occurrence of temperature and ozone exceedances over Europe. To do so we propose a copula-modeling approach in order to (i) model the dependence structure between high ozone concentrations and high temperatures under blocking and non-blocking conditions and (ii) quantify the impact atmospheric blocking on the joint probabilities of exceedances derived from the copulas. In the context of multivariate processes that may lead to compound events, the application of copula-based probability has been widely used recently (e.g., Hao et al., 2018, and references therein). Copula-based methods have been

extensively applied in hydrological extremes (Salvadori and Michelle, 2010; Hao et al., 2018) and provide a flexible way of construction for a joint distribution with arbitrary marginal distributions (AghaKouchak et al., 2014). Copulas describe the dependence between random variables (Nelsen, 2006) and, besides characterising the overall dependence structure, certain copula families allow to measure the upper tail dependence, which is particularly important to assess extreme events (Zscheischler and Seneviratne, 2017; Serinaldi, 2016). Therefore, with the main goal of estimating the effects of atmospheric blocking

on the relationship between ozone and temperature, we apply a copula-based approach that allows us to quantify the influence of atmospheric blocking on the upper tail of the joint distribution of ozone and temperature.

## 2   Data and methods

Daily maxima of the 8-hour average (MDA8O$_3$) of ozone concentrations were extracted from the European Environment Agency's air quality database (AirBase) (https://www.eea.europa.eu/data-and-maps/data/) during the period 1999-2015 focus-

ing on the ozone season that spans from April to September. The ozone season refers to a period of time in which surface ozone levels typically reach the highest concentrations (e.g., Schnell and Prather, 2017; Otero et al., 2018). A total of 300 background monitoring stations, including rural, urban and suburban, with altitude $< 1000$ m and with at least 75% of valid data available for each ozone season, were used. The number of stations for which measurements are available vary greatly in space, with the major density of stations being over central Europe. However, a representative number of stations over northern and southern

Europe are also included (Fig. S1).

The daily maximum temperature was derived from the 6-hourly of the 2 m-temperature values extracted from the ERA-Interim (Dee et al., 2011) reanalysis of the European Center for Medium-range Weather Forecasts (ECMWF) for the same

period 1999-2015. The temperature dataset was available at $1° \times 1°$ regular (latitude/longitude) resolution. The daily 500 hPa geopotential height (Z500) field was obtained from the ERA-Interim reanalysis at a coarser horizontal resolution of $2.5°x 2.5°$ (latitude/longitude), which is appropriate to characterise large-scale atmospheric circulation.

## 2.1 Blocking detection

A 2-dimensional blocking index (BI) derived from daily Z500 was used to identify instantaneously blocked grid points. This blocking index is calculated according to the one-dimensional index proposed by Tibaldi and Molteni (1990) but expanded to every latitude and longitude (Scherrer et al., 2006). Similar to Barnes et al. (2014), we apply a spatio-temporal filter that allows the exclusion of small-scale and short-term blocking situations accounting for large-scale and persistent systems between $35°$ and $80°N$. Thus, we select contiguously blocked regions with a minimum zonal and meridional extension of $15°$ and an area of at least $1.5 \times 10 \ 10^6 \ km^2$. A persistent blocking event is considered if the duration of the blocking system lasts a minimum of four days. In addition, the tracking algorithm includes possible merging and splittings of the blocking event in time by adopting a blocking overlap area criterion of $7.5 \times 10^6 \ km^2$ between two consecutive days and a maximum distance between blocking centers of 1000 km (Schuster et al., 2019).

The blocking index was calculated using the 'Free Evaluation System Framework' (Kadow et al., 2021), which is a framework for scientific data processing designed for atmospheric applications that includes (among other features) software for the calculation of the BI; details about this method are given in Richling et al. (2015).

## 2.2 Joint distribution analysis

Recently, copula-based approaches have become very popular to assess interrelations between several random variables (Ribeiro et al., 2019; Salvadori et al., 2016; Hao et al., 2018). A copula is a joint distribution function in which the marginal distributions are independent of the dependence structure and can be modeled separately (Nelsen, 2006). For two random variables $X$ (temperature) and $Y$ (MDA8O$_3$) with marginal distributions $F_X(x) = Pr(X \leq x)$ and $F_Y(y) = Pr(Y \leq y)$, respectively, a copula function allows to construct their joint cumulative distribution as follows:

$$F_{XY}(x,y) = C(F_X(x), F_Y(y)) = C(u,v) \tag{1}$$

where $F_{XY}$ is the joint distribution function of $X$ and $Y$, C is the copula function and u = $F_X$ and v = $F_Y$ are the uniformly distributed marginals. According to Sklar's theorem, if the marginal distributions are continuous, then the copula function C is unique (Sklar, 1996). The main advantage then of using copula functions is the flexibility to model the dependence between multiple random variables that follow arbitrary univariate marginal distributions. For each station, we use bivariate copulas to model the dependence between temperature and ozone and estimate their joint probability distribution under two different synoptic situations: 1) when there is a co-located block in the same location of MDA8O$_3$ and temperature (i.g BI=1), 2) without the presence of blocking (i.e. BI=0). We fit a total of four commonly used copulas: t-student (from the Archimedean family), Clayton, Gumbel and Joe (from the elliptical family) (Table 1). The Archimedean copulas are able to describe asymmetric

tail behaviour, while elliptical copulas capture symmetric dependence (Tilloy et al., 2019). Among the different copulas, we selected the structures that are able to capture tail dependence; Gumbel and Joe copulas model upper tail dependence, while Clayton can capture lower tail dependence. (Salvadori and Michelle, 2010). The t-student copula allows dependence in both, upper and lower tails. Before modeling the joint probability distribution, we fit the most appropriate marginal distribution for both temperature and MDA8O$_3$, including Gaussian, gamma, Weibull and lognormal distributions. The parameters for the marginal distribution were obtained by the maximum likelihood method separately for each site. The marginal distributions were selected using the Kolmogorov–Smirnov goodness-of-fit test. Then, for each station and synoptic case (BI=1 and BI=0), the bivariate copulas were selected based on the Akaike's Information Criteria (AIC) (Akaike, 1974) and the copula parameters were estimated via maximum likelihood estimation (MLE). The copula analyses were carried out with the VineCopula and the copula R packages (Schepsmeier et al., 2016; Hofert et al., 2020).

**Table 1.** Equations of the copula functions, where u and v are univariate variables (uniform distributed), $\theta$ and $\rho$ are the dependence parameters and df is the degree of freedom.

| Copula Family | Function | Parameter range | Upper tail | Lower tail |
|---|---|---|---|---|
| Gumbel | $C(u,v) = \exp(-[(-lnu)^\theta + (-lnv)^\theta)]^\theta$ | $[1,\infty)$ | Yes | No |
| Clayton | $C(u,v) = (u^{-\theta} + v^{-\theta} - 1)^{-1/\theta}$ | $(0,\infty)$ | No | Yes |
| Joe | $C(u,v) = 1 - ((1-u)^\theta + (1-v)^\theta - (1-u)^\theta(1-v)^\theta)^1/\theta$ | $[1,\infty)$ | Yes | No |
| t-student | $C(u,v) = \int_{-\infty}^{tdf^{-1}(u)} \int_{-\infty}^{tdf^{-1}(v)} \frac{1}{2\pi\sqrt{1-\rho^2}} exp(1 + \frac{u^2+v^2-2\rho uv}{df(1+\rho^2)})^{-\frac{df+2}{2}} dudv$ | $-1 \leq \rho \leq 1; 1 \leq df$ | Yes | Yes |

The copula models were used to assess the relationship between temperature and ozone exceedances under blocking (BI=1) and non-blocking (BI=0) conditions by constructing the corresponding joint probability distribution (P($X \leq$ x, $Y \leq$ y)). Apart from the general dependence structure, some copulas can measure the dependence of the extremes through the tail dependence parameter ($\lambda_u$) (Nelsen, 2006). As linear or rank dependence measures might not be accurate when focusing on extremes (Hao and Singh, 2016), we have further assessed the upper tail dependence of ozone and temperature extremes derived from the copulas under blocking (BI=1) and non blocking (BI=0). We estimate the probability of a compound event at each station, in which Tmax and MDA8O$_3$ exceed the 95th percentile of their respective distribution. It is important to note that the 95th percentile for each variable is calculated over the whole distribution of the ozone season, from April to September, for the period of study, i.e., 1999-2015. Therefore, the compound extremes at each station are defined based on a relative threshold value, defined for each station as a function of the 95th percentile over the whole distribution (i.e., including non blocked and blocked days) of the corresponding variable (i.e., temperature and ozone). This choice was made in order to quantify the impact of blocks on the probability of exceedances, for which the definition of extreme changes depending on the station.). Precisely, the use of absolute thresholds allows us to quantify the impact of blocks on the probability of exceedances. The probability of exceedances over a certain multivariate threshold was examined based on three different hazard scenarios described by

the following joint and conditional joint probabilities, which can be expressed using copula notation (see further details in Serinaldi, 2015):

$$P_{AND} = P(U > u \cap V > v) = 1 - u - v + C(u,v) \tag{2}$$

$$P_{OR} = P(U > u \cup V > v) = 1 - C(u,v) \tag{3}$$

$$P_{COND} = P(U > u | V > v) = (1 - u - v + C(u,v))/(1 - u) \tag{4}$$

The probabilities in equations 2 and 3 have been widely applied in the literature to assess compound extremes (Zscheischler and Seneviratne, 2017; Hao et al., 2018). Equation 2 represents the scenario in which both variables temperature (Tmax) and ozone (MDA8O$_3$) exceed the 95th percentile, while equation 3 considers a situation wherein the events occurred when either temperature or ozone or both exceed their respective threshold (95th). As blocks normally lead to persistent positive surface temperature anomalies during summer over Europe (Pfahl and Wernli, 2012), it is of interest to evaluate the influence of blocking on the probability of ozone exceedances given high temperatures, which is assessed in the $COND$ scenario.

To quantify the significant impact of blocks on the compound ozone and temperature events, we estimated the differences between the probabilities derived from the copulas (i.e., $\Delta P = P_1 - P_0$). Then, we assessed whether the difference between the probabilities when BI=1 and BI=0 are significantly different from zero. To do so, we apply a bootstrap procedure for each probability scenario (i.e., $AND$, $OR$, $COND$) in which we drew 100 bootstrapped samples and derived the respective probabilities $P_1$ and $P_0$ when BI=1 and BI=0, respectively. For the null hypothesis ($H_0$), there is no difference between the probabilities obtained from the cases BI=1 and BI=0, while the alternative hypothesis indicates that the probability of an extreme event conditioned to a blocking situation is significantly different from the probability under non blocking conditions.

## 3 Results

We begin our analysis by examining the frequency of atmospheric blocking over Europe for the period of study. Afterwards, we analyse the effect of atmospheric blocking separately for MDA8O$_3$ and Tmax in order to analyse the impact of blocks on the margins. In addition, we examine the influence of blocking on the statistical correlations between MDA8O$_3$ and Tmax before modeling the dependence structure with the copula-approach. Thus, we first provide exploratory analysis and then continue with the risk assessment through the joint probability derived from the copulas.

### 3.1 Impact of atmospheric blocking on ozone and temperature

For the period of study (1999-2015) a total of 3111 days were analysed during the ozone season (April-September). The blocking frequency (% of days) ranges between 5% in the southern latitudes (30-45° N) and 14% in the northern latitudes (60-

70° N) (Fig. S1). In central Europe, where the density of stations is higher, the frequency of blocked days is $\sim 8\%$. Typically, blocking presents a well-established climatology in terms of frequency in the North Hemisphere, being more frequent in winter and spring, and less frequent in autumn (Barnes et al., 2014; Wollings et al., 2018). During summer, blocking events have shown a tendency to occur at high latitudes over continental areas (Barriopedro et al., 2010). In contrast with previous related studies analysing the seasonal responses of air pollution to blocks (Ordóñez et al., 2017; Garrido-Perez et al., 2017), our study focuses on the whole ozone season during which the compounding effect ozone and temperature is particularly relevant for human health (Hertig et al., 2020). Moreover, atmospheric blocking events are likely to have a major impact due to their connection with heatwaves in spring and summer (Brunner et al., 2017).

We start by examining the individual impacts of blocks on the anomalies of ozone and temperature, in order to establish a comparison of anomalies across different stations. MDA8O$_3$ anomalies were calculated as the difference between MDA8O$_3$ values and the average of MDA8O$_3$ over all days in April-September during the period of study, i.e., 1999-2015. This average is obtained individually for every station. Tmax anomalies were similarly calculated with respect to the average value over all Tmax values from April-September during the same period. It is important to highlight that all calculations were applied separately for each station, and therefore, the number of blocking days might differ across the different stations. Figure 1 illustrates the composites of the anomalies of both, MDA8O$_3$ and Tmax, during blocking days (BI=1). In general, most of the stations show positive anomalies of MDA8O$_3$ under blocking days (Fig.1, left). The strongest positive anomalies $> 30 \, \mu\text{gm}^{-3}$ are observed over the south of Germany, north-east of France and north-west of Italy, while weaker anomalies are found over Scandinavia, west of the UK and north of Spain. Similarly, Ordóñez et al. (2017) reported strong positive anomalies over large areas of central and northern Europe in spring and summer respectively. In the case of temperature anomalies, blocks led to positive anomalies of Tmax over all of the stations included in this study (Fig. 1 right). The largest values of temperature anomalies ($>7°$C) are observed over the central and western stations (north-east of France, Austria and south of Germany). This is consistent with the radiative heating due to enhanced clear sky conditions over continental areas under atmospheric blocking conditions, especially in summertime (Brunner et al., 2017).

We further examined the impact of blocking on individual extremes of ozone and temperature. To this end, we do not work with anomalies but, instead, fix a threshold for MDA8O$_3$ as well as Tmax. The absolute values for these thresholds vary among stations. A transparent way to set these thresholds are quantiles, i.e., values with a specified non-exceedance probability. For each station, we use the 0.95-quantile (or 95th percentile) from the sample restricted to April to September and thus get individual thresholds for all stations reflecting their local climatology. Thus, we define days with MDA8O$_3$ exceeding this threshold as extreme days. Then, we obtain relative frequencies by dividing the number of extreme days with a simultaneous blocking by the number of total days in the data set restricted to April to September (i.e., $\hat{p} = extremes with blocking/3111$). A similar approach was applied in Ordóñez et al. (2017) that calculated the percentage of blocking days with MDA8O$_3$ values above the 90th percentile. It must be noted that the spatial variability of high levels of ozone is very heterogeneous (Fig. S2). The 95th percentile of MDA8O$_3$ exceeds the European target value of MDA8O$_3$ (120 $\mu\text{gm}^{-3}$, EEA, 2019) in a large number of stations in central and south Europe. Only in the case of northern stations (UK and Scandinavia), MDA8O$_3$ would not often exceed the mentioned target value (Fig. S2).

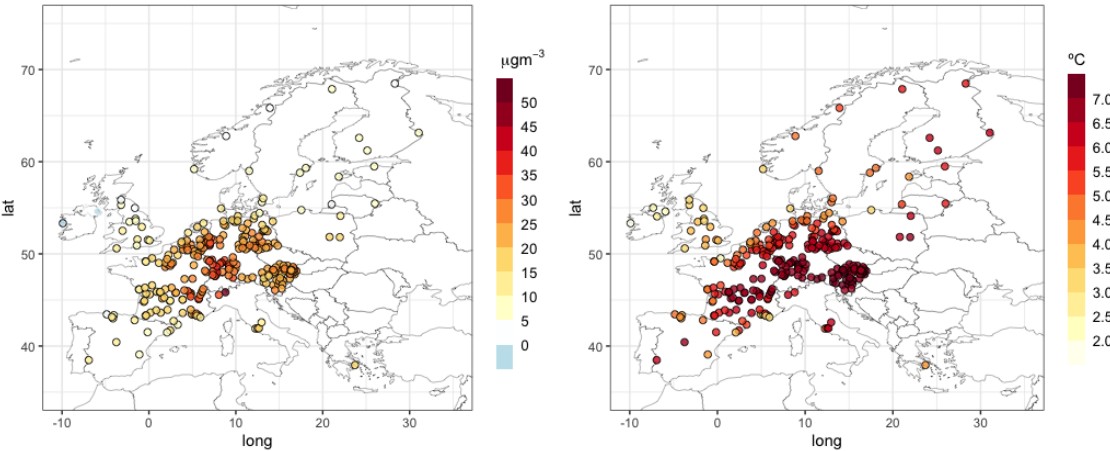

**Figure 1.** Anomalies of MDAO3 (left) and Tmax (right) for blocking days (i.e., BI=1). Anomalies of MDAO3 were calculated with respect to the MDAO3 concentrations over the whole period 1999-2015 during the ozone season, April-September. Similarly, anomalies of temperature were obtained with respect to the temperature over the whole period of study (as for MDAO3). Black contour indicates statistically significant anomalies at the 95% confidence level of a two-sided t-test.

The same procedure using the 95th percentile was applied for identifying days of Tmax exceedances and days above the 95th percentile of the Tmax (Fig. S2) were classified as exceedances. We acknowledge that, in the case of Tmax, the number of exceedances above the 95th percentile might be not equally distributed across the ozone season (i.e., this threshold is more likely to be exceeded in July and August than it is in April and September). While this could be corrected by either using a threshold that varies seasonally or by removing the seasonal trend in the data, we would like to stress that the main goal of this

study is to quantify the impacts of blocking on the upper-tail dependence between $MDA8O_3$ and Tmax over the entire ozone season. Our main interest is in the physiological effects of such compound events, for which only absolutely high temperatures (as they tend to occur in July or August) are relevant. Similar to other studies (e.g., Schnell and Prather, 2017), we use the 95th percentile over the period between April September. A lower threshold, e.g., the 90th percentile would lead to many temperature values not being physiological relevant; a higher threshold, e.g., the 99th percentile, on the other hand, would

lead to a strong reduction in the data available for the subsequent copula modeling. The 95th percentile-based definition to examine the individual impacts of blocks on $MDA8O_3$ and Tmax it is also justified to be consistent with the joint probability analysis, for which the 95th percentile is applied for the risk assessment (see below). Moreover, earlier studies used a similar threshold percentile-based definition to assess the links between temperature extremes and atmospheric blocking (Pfahl and Wernli, 2012).

Figure 2 illustrates the frequency of blocked single extremes of ozone and temperature (i.e., the percentage of exceendances of $MDA8O_3$ and Tmax with respect the total number of blocked days). More than 40% of $MDA8O_3$ extremes coincide with blocked days over most of the central stations. The frequency of blocked days of $MDA8O_3$ extremes is generally lower in the southern stations. The percentage of Tmax extremes coincident with blocks increases northwards and eastwards, which is

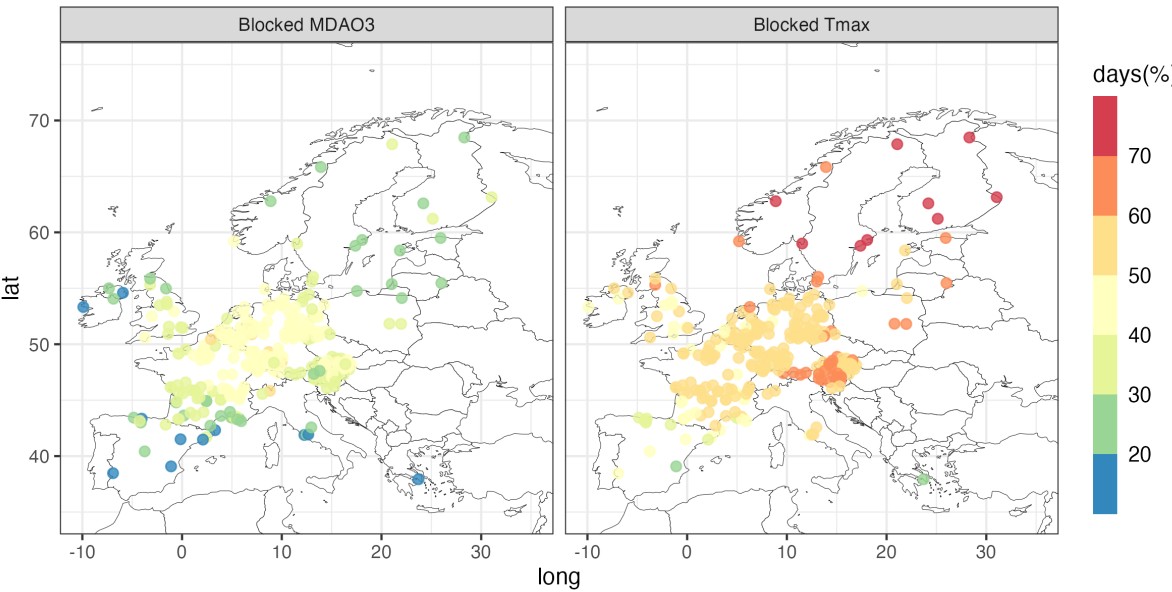

**Figure 2.** Percentage of days with MDA8O3 (left) and Tmax (right) exceedances (>95th) that are blocked days.

consistent with subsidence processes and the clear-sky radiative forcing associated with summer blocking events (Pfahl and
Wernli, 2012; Sousa et al., 2018). Moreover, as stated above, the strong seasonal variability of the blocking activity in the
Northern Hemisphere must be noted with a reduced number of occurrences late in summer and autumn but being considerably
more frequent in eastern Europe than in the Euro-Atlantic region (Barriopedro et al., 2006). This pattern is also reflected in our
results that show the largest number of blocked temperature extremes north and eastwards.

To investigate the impact of blocks on the relationship between MDA8O$_3$ and Tmax, the Kendall's tau coefficient ($\tau$) was
calculated during blocking and non blocking days as well as the difference between the correlations obtained when BI=1 and
BI=0 (Fig.3 a and b). The correlations are weaker under non-blocking days (BI=0) and a few number of northern stations show
negative values (Fig. 3 a). In general, the dependence between MDA8O$_3$ and Tmax is higher under the influence of blocks.
The positive differences between the correlation values (Fig.3 b) clearly reflect the strong impact of blocks at most of the
stations. The largest differences are found over the north-west of Europe. Consistent with previous work, the central stations
show the strongest relationship between ozone and temperature (Otero et al., 2016), which significantly increases when BI=1
with the largest correlation values (>0.6). Blocks seem to have a great influence over the north-west of Europe in particular,
the UK, north of France and Benelux (i.e., Belgium, Netherlands and Luxembourg), where the correlations are higher and
mostly positive under blocking conditions, while negative correlations are found under non blocking. A similar pattern was
found when calculating the correlations for the respective extremes based on the 95th percentile that showed the strongest
relationship under the influence of blocks over a large number of stations of France, Germany and the UK (Fig. S3). The

impact of blocks in the relationship between $MDA8O_3$ and Tmax is smaller in the south and north-east, which is reflected by non significant and weaker correlations that show similar magnitude values when BI=1 and BI=0.

The results from the individual impacts of atmospheric blocking on $MDA8O_3$ and Tmax are consistent with previous studies that showed the impacts of blocking on ozone (Ordóñez et al., 2017) and the association between blocking and temperature (e.g., Sousa et al., 2018; Sillmann et al., 2011; Pfahl and Wernli, 2012). Consistent with these works, we found notable spatial differences, with the largest blocking effects on the north western and central stations and weaker impacts on the southern stations. Ordóñez et al. (2017) showed that subtropical ridges, an extension of the sub-tropical high pressure belt extending northwards (Sousa et al., 2018), had a major impact on surface ozone in the central-south European sectors, especially in summer, while blocks showed a stronger impact in central and northern Europe in spring and summer, respectively. Nevertheless, they pointed out that the influence of ridges for the build-up of ozone pollution is not as clear as in the case of blocks, and its impact is more sensitive to the location. Using a similar catalogue to detect blocks and sub-tropical ridges, Sousa et al. (2018) showed that blocks play an important role in warm temperature anomalies in spring and summer over central Europe, while the impact is generally lower over south, mostly due to the position of the block (Sousa et al., 2018). It must be noted that our detection method only focuses on blocks and, unlike the cited works, sub-tropical ridges were not included for this analysis. Despite this, our results are in good agreement with their findings and they also point out the spatial variability of the blocking effects on both, $MDA8O_3$ and Tmax. Previous work has shown a strong effect of $NO_x$ levels on the temperature sensitivity of ozone (Pusede et al., 2014; Coates et al., 2016; Otero et al., 2021). Due to the relatively short atmospheric lifetime of $NO_x$ (order of hours), the combined effect of blocking and high temperature on ozone would be larger in areas close to strong $NO_x$ sources, such as large urban areas. Thus, we might anticipate spatial differences in the impact of blocks on the compound extremes of ozone and temperature and their joint distribution.

## 3.2 Copula results

In the previous section, we have shown, separately, the effects of atmospheric blocking on both, $MDA8O_3$ and Tmax as well as the influence on their relationship through the correlation coefficients. Here, we present the results from the copula modeling analysis which allow us not just to confirm the impacts of blocks on the structure dependence between $MDA8O_3$ and Tmax previously shown, but also to quantify the impacts of blocks on the compounding effect of both variables by estimating the joint probability of exceedances.

Among the different types of copulas presented in the literature, a total of four copulas (Table 1) were tested to find the most appropriate fit that characterises the relationship between $MDA8O_3$ and Tmax at each station. Our copula choice was mainly motivated by their ability to represent joint tail dependence (upper and/or lower). After modeling the dependence between both variables, when BI=1 and BI=0 separately, we quantify the effect of atmospheric blocking on compound extremes of ozone and temperature through the differences between probabilities derived from the cases mentioned above (BI=1 and BI=0) for each probability scenario. The impact of blocks on the joint behaviour between $MDA8O_3$ and Tmax is reflected in the selected copula (Fig. S4). When BI=0, a large number of stations are characterised by an asymmetric dependence structure as it is the case of Joe and Gumbel copulas. The Gumbel copula is also selected in a number of stations when BI=1, but, in this case, the

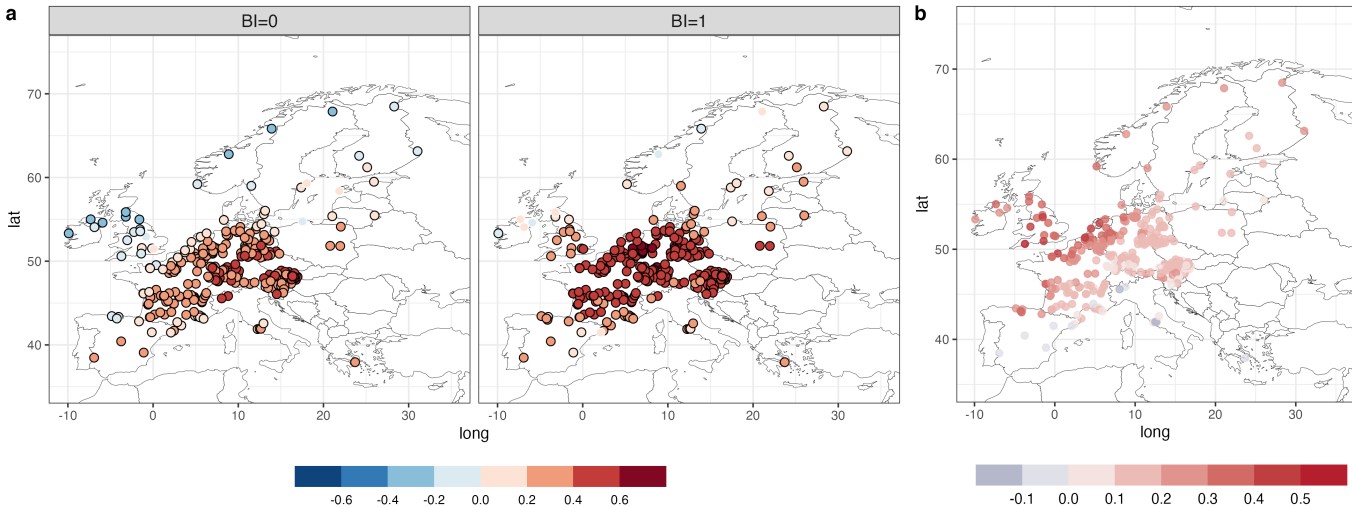

**Figure 3.** a) Spatial distribution of the Kendall's correlation coefficient between MDA8O3 and Tmax under non-blocking contidions (i.e., BI=0) and blocking conditions (i.e.,BI=1). Black contour indicates statistically significant anomalies at the 95% confidence level of a two-sided t test. b)Differences between the correlation values obtained when BI=1 minus the correlations when BI=0.

280 t copula is representative of a major number of stations. Contrary to the Gumbel and Joe copulas, the t copula belongs to the Elliptical and radially symmetric copulas, but captures dependence in the extremes in both, lower and upper tail (Nelsen, 2006). We further investigated the influence of blocks on the upper tail dependence parameter,$\lambda_u$, obtained from the chosen copulas. The $\lambda_u$ measures the tendency of concurrent extremes of MDA8O$_3$ and Tmax exceeding the 95th percentile. According to Fig. 4, the strongest upper tail dependence occurs under the influence of blocks over north-west and central Europe (e.g., the
285 UK, France, Benelux and north of Germany). The impact of blocks is particularly noticeable over the UK and Benelux with an increase in the dependence of extremes when BI=1, which is well observed when plotting the differences between the values of $\lambda_u$ obtained for both cases (Fig.4 b). This pattern is in agreement with the relationship obtained by Kendall's $\tau$ (Fig.S3) that shows a stronger relationship of extremes under the influence of blocking conditions.

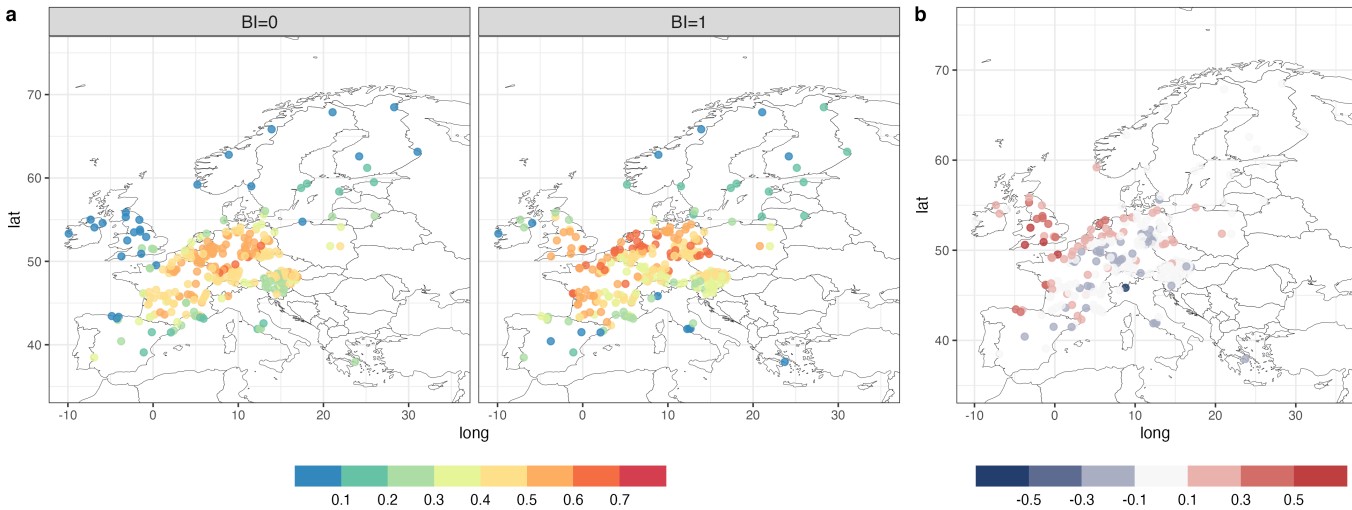

**Figure 4.** a) Spatial distribution of the upper tail dependence parameter derived from the copulas when BI=0 and BI=1. b) Differences between the values of the upper tail dependence parameter when BI=1 minus the values when BI=0.

We use three hazard scenarios ($AND$, $OR$, $COND$) to quantify the impacts of blocks on compound extremes of MDA8O$_3$
and Tmax. The probabilities associated with each type of hazard scenario are defined based on the domain where they are estimated and the critical region related to the probability type (see Fig. 5 for an illustrative example as shown in Serinaldi, 2016).

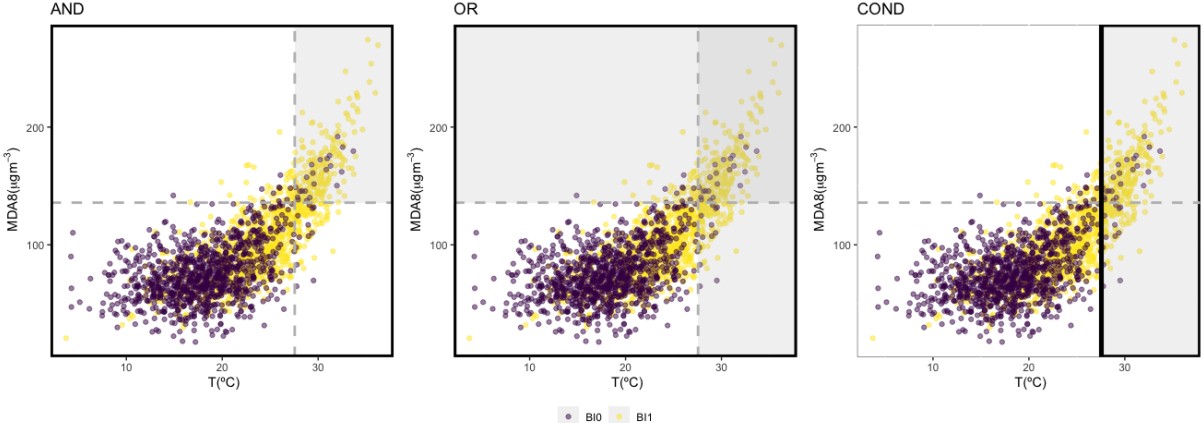

**Figure 5.** Illustrative example with the probability scenarios used in the study: AND (left), OR (middle), and COND (right). Bold black boxes identify the domain where each probability is estimated and the grey areas represent the critical regions associated with the corresponding probability. The legend colors correspond to days with blocking (BI=1, yellow) and without blocking (BI=0, purple).

We start analysing the impacts of blocks in the probability of concurrent events of high ozone pollution and hot days using the scenario $AND$ (Fig. 6 a and b). While there is a very low probability in the co-occurrence of extremes when BI=0 ($< 2.5\%$),

the presence of blocking generally increases the probability of compound events of MDA8O$_3$ and Tmax (Fig. 6 a). Under blocking condtions (BI=1), the probability of co-occurrent extremes is $\sim 20\%$ in most of the stations over central Europe, and $\sim 10\%$ over the UK (Fig.6 a). The probability of occurrence of compound events of MDA8O$_3$ and Tmax significantly increases by more than 18% in a large number of stations over Europe, as shown by the $\Delta P$ (Fig. 6 b). Despite ozone concentrations being generally lower over the UK (Fig.S2) compared to other regions, it is interesting to note that blocks seem to play a

significant role in the compounding effect ozone and temperature over the UK. Kalisa et al. (2018) analysed the influence of heatwaves on air pollution in the UK, specifically Birmingham, and found that ozone levels increased by more than 50% with high temperature. Here, we consistently show the combined effect ozone and temperature. Our results also indicate that such combination mainly occurs under blocks, which might be due to the clear-sky radiative forcing, as pointed out by earlier work, and subsidence processes associated with the anticyclonic circulation (Brunner et al., 2017; Pfahl., 2014). The stations over the

north-eastern and south-western stations (i.e., Scandinavia and Spain, respectively) exhibit the lowest probability of compound of extremes of MDA8O$_3$ and Tmax. As shown in Fig. 4, those stations are characterised by low or null upper tail dependence, which already indicates a weak relationship between the extremes. In addition, the distinct response of heatwaves to blocking found in northern and southern Europe is noteworthy, especially in summer (Brunner et al., 2017; Sousa et al., 2018). An increasing frequency of heatwaves linked to blocks has been observed over northern Europe in summer, while an opposite

response has been seen in southern Europe (Sousa et al., 2018). Therefore, one could expect a smaller impact of blocks on the compounding effect of ozone pollution and high temperatures in the case of the southern regions. Our results are in agreement with the study carried out by Hertig et al. (2020) that found a lower number of compound ozone-heat wave events in Portugal compared to the compound identified in Germany (Bavaria).

We examine the $OR$ scenario under the assumption that blocks might enhance the probability of either high ozone pollution

levels or hot temperatures, both being relevant for health impacts (Analitis et al., 2014; Bell et al., 2004). As shown in Fig. 6 (c), the probability obtained for the $OR$ scenario is considerably higher when BI=1, reflecting the strong impact of blocks on single extremes events. Atmospheric blocking conditions enhance the probability that either MDA8O$_3$ or Tmax exceeded the 95th percentile more than 40% in a large number of stations mostly concentrated in Germany, Austria and the east of France. For the rest of the stations, the probability of extremes in the $OR$ scenario increases by 20-30% under blocking conditions (Fig.6

320 d). Consistent with previous works that showed the strong association of warm temperature extremes and blocking (Pfahl and Wernli, 2012; Brunner et al., 2017), as well as the impact of blocks on ozone pollution over some European sectors (Ordóñez et al., 2017), our results show the increasing probability of temperature $OR$ ozone pollution extremes under atmospheric blocking.

From a risk assessment perspective, the scenario $COND$ is also of interest as it quantifies the impact of blocks of ozone

pollution extremes events conditioned on high temperature. For the $COND$ probability, both, the computation domain (i.e., the joint space where the probability of exceedances is calculated) and the critical region (i.e., the region of exceedances of ozone conditioned by temperature) evolve when moving along higher temperatures; then, the probability is computed over a reduced

subset (e.g., conditioned on temperature extremes) (see Fig.5 and Serinaldi, 2016, for further details). As illustrated in Fig. 6 (e and f), blocks generally enhance the probability of extremes of ozone pollution conditioned on temperature exceedances.

Blocks significantly influence the compound events in the stations over north-west and central-east of Europe that show positive and large values of $\Delta P$ (Fig. 6 f), suggesting a higher probability of ozone pollution extremes when temperature exceeds the 95th. In particular, blocks lead to an increasing probability of ozone extremes given high temperatures in the UK (>40%). In a few number of stations over south and north-eastern Europe, blocks did not show a significant influence in the conditional probability of extremes, with low and non-significant values of $\Delta P$. For some of these stations, the copula selected when BI=1

is the Clayton copula (Fig. S4), which indicates a greater probability of joint extreme low values (lower tail dependence), but not in the upper tail, as shown in Fig. 6 (e and f). Thus, the presence of blocks is not relevant for ozone pollution exceedances that seem to occur independently of temperature extremes. In such situations, high ozone levels are less likely to be due to the enhanced local ozone production from locally emitted precursors that comes with higher temperatures (Coates et al., 2016), and more likely to be due to residual ozone left over from previous episodes of enhanced local ozone production (Haman et al.,

2014), or long-range transport of ozone produced elsewhere (Lupaşcu and Butler, 2019).

The results from the joint probabilities derived from the copulas pointed out notable spatial differences that were consistent with the analysis presented above. The impacts of blocks on the joint probabilities corresponding to the $AND$ and $OR$ scenarios is significant at all stations, with a major effect (in terms of the magnitude of $\Delta P$) in those located in central Europe. The smallest impact was found at the southern and north-eastern stations for the conditional case, $COND$, that did not show a

345 significant impact of blocking. Despite not considering the sub-tropical ridge in our methodology, the results from the copula analysis are in line with previous studies, which showed the spatial variability in the impacts of blocking.

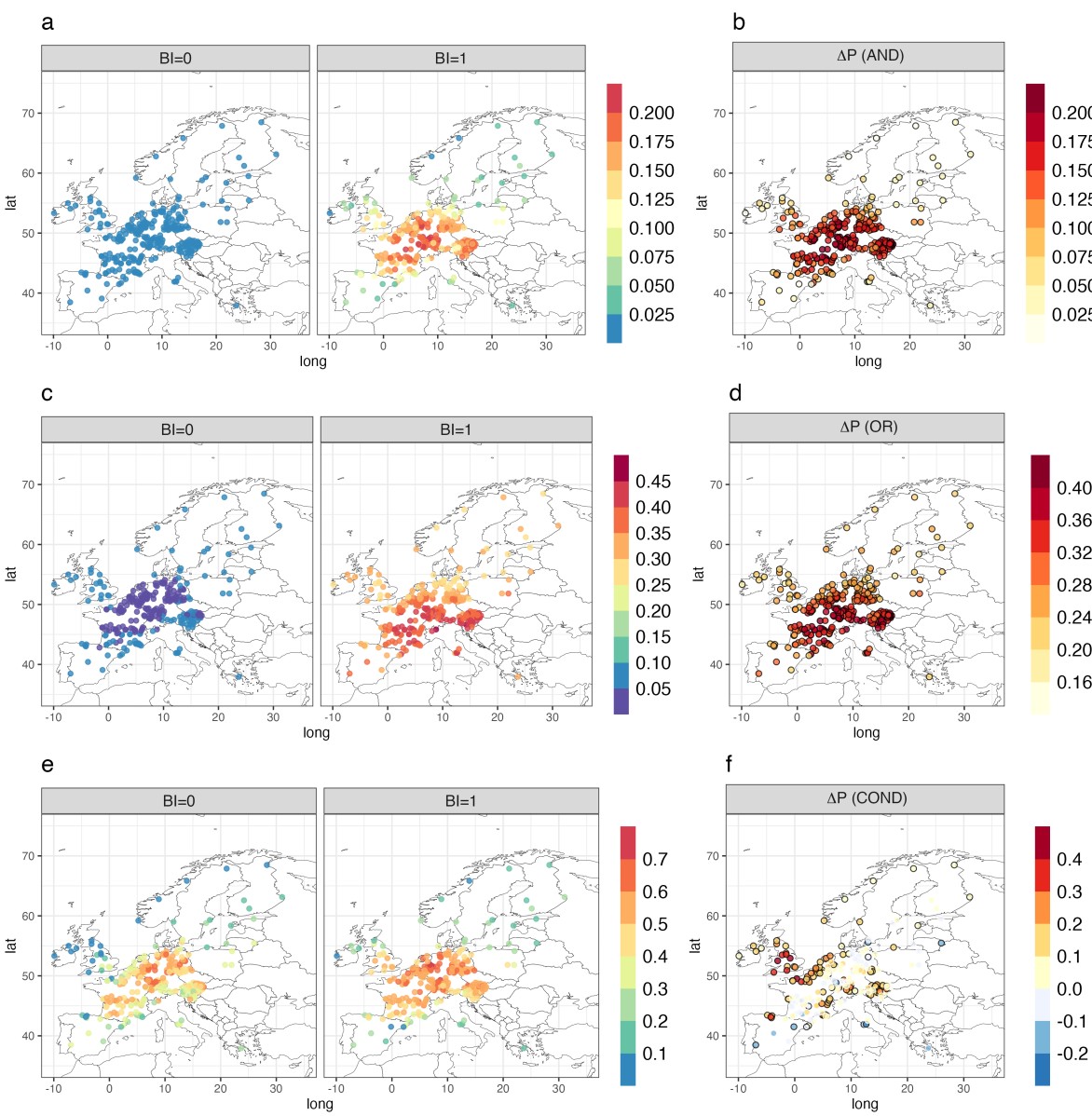

**Figure 6.** Probability scenarios AND (a and b), OR (c and d) and COND (e and f) derived from the copula analysis when BI=1 and BI=0 (a, c and e). ΔP (b, d and f) shows the difference between the probabilities when BI=1 and BI=0. Black contours in b), d) and f) represent locations with statistically significant differences at the confidence level of 95%.

## 4  Discussion and conclusions

The present study has assessed the influence of atmospheric blocking on the dependence between daily maximum of 8h average ozone (MDA8O$_3$) and daily maximum temperature (Tmax) for the period 1999-2015 during the ozone season (April-September). A total of 300 monitoring stations distributed over Europe were included. First, we examined the blocking influence on single extreme events of ozone pollution and temperature, defined on the basis of the 95th percentile of their respective distribution at each station. Using a copula-based approach, we evaluated the impacts of blocks on compound ozone pollution and temperature events taking into account their dependence. For each station, the dependence between ozone and temperature was modelled independently under blocking (BI=1) and non blocking (BI=0) conditions. The selected copulas described the dependence structure and the joint behaviour of ozone and temperature. We investigated the impacts of blocks on the risks of compound ozone and temperature events under three different hazard scenarios of probability: $AND$, $OR$ and $COND$, which are commonly used to study multivariate events.

Our results showed that, during the ozone season, more than 40% of ozone exceedances (>95th) are coincident with blocked days over the central stations (including Germany, east of France and Benelux). The rest of the stations showed a lower frequency ($\sim 25\%$) of ozone exceedances during blocking conditions. The frequency of temperature extremes is larger than ozone extremes under blocking conditions and, on average, 55% of hot days occur under blocking conditions. The highest frequency is observed in northern Europe (Scandinavia) with more than 70% of temperature-blocked extremes, while the lowest frequency is observed in southern Europe. It is worth noting that in the case of temperature, the number of exceedances above the 95th percentile of the total distribution (i.e., April-September) might not be equally distributed throughout the ozone season. However, we consider that the use of a fixed 95th percentile for the whole distribution to define individual extremes is also consistent with the 95th percentile threshold used to estimate the joint probabilities of exceedances derived from the copulas. Moreover, our results were in agreement with the literature that showed similar patterns of temperature-blocked exceedances (Brunner et al., 2018; Sousa et al., 2018).

The analysis of the dependence between ozone and temperature revealed that atmospheric blocking is of key importance in some regions that showed a strong relationship between ozone and temperature under blocking conditions (e.g., central and eastern Europe). In particular, we found a great impact over the stations in the UK and Benelux where the blocks lead to positive and higher correlation values, while a weaker relationship is observed under non-blocking conditions. The copula-based approach confirms the dependence between ozone and temperature under the influence of atmospheric blocking. Moreover, the copulas showed that blocks have a major effect on the upper tail dependence in some stations over the UK, north-west and west of France, Benelux and north of Germany, which suggests that compound ozone and temperature extremes are highly associated and influenced by atmospheric blocking.

Overall, we found that blocks enhanced the probability of occurrence of compound ozone and temperature extremes in a large number of stations included in this study. Our results showed that blocking significantly increased by $\sim$ 15%-20% (i.e., $\Delta P > 0.15$) the probability of co-occurrent ozone and temperature exceedances at the stations over central, north-west and east of Europe. In fact, the probability of combined ozone and temperature extremes under non blocking conditions is rather

small everywhere ($P_0 < 0.025$). Blocks significantly increase the probability that either ozone or temperature (or both) exceed the 95th percentile. The highest probability values are observed over central and eastern stations in which blocking increases the probability of extreme events in ozone or temperature by more than 35%. The analysis of the joint distribution considering the conditional hazard scenario ($COND$) showed a smaller impact of blocks in some stations where the probability of ozone pollution extremes conditioned on high temperature did not show significant differences in terms of magnitude under non-blocking conditions. However, we found a significant increase in the conditional probability over the north-west stations and a slight increase over the central-east stations. This suggests that, over such regions, ozone extremes tend to occur conditioned on high temperatures which are strongly connected with atmospheric blocking. This is likely due to the position of the block (i.e., the location of the center of the identified block) during the ozone season covering spring and summertime when the increased solar radiation lead to warm temperature in the blocked regions (Brunner et al., 2017; Sousa et al., 2018), which can also explain the high levels of ozone pollution in the blocked regions. As described in section 2, we used a blocking detection algorithm based on the instantaneous blocking index developed by Tibaldi and Molteni (1990) and applied an additional spatio-temporal filter. It must be noted that, unlike earlier studies, we considered the blocks within the Atlantic and European sectors, mainly motivated by the location of the stations, but we did not explicitly analyze other properties of blocks, such as blocking centre, blocking duration, or blocking extension, which might have an effect on the compounding response of ozone and temperature. Future directions from this work might assess the role of the blocking properties on the probability of co-occurrence of temperature and ozone extremes.

Our study showed a clear influence of blocks in local compound ozone and temperature extremes over a large number of stations. Blocks have a significant impact over the central regions, where peaks of ozone pollution usually exceed the European target value of $120 \ \mu\mathrm{gm}^{-3}$ (set for the protection of human health, EEA, 2019) and warm temperature extremes are strongly connected with atmospheric blocking (Brunner et al., 2017). Ozone levels are normally lower over north-western Europe (e.g. , the UK) as well as temperature compared to the rest of the stations (Fig. 2); however, our findings showed that blocking leads to an increased strength of the general dependence between ozone and temperature, particularly in the tail dependence of extremes. This points out that blocks have a significant impact in the compounding effect of ozone and temperature over north-western Europe, leading to higher health risks.

As discussed in the introduction, atmospheric blocking can lead to extreme weather conditions (Sillmann et al., 2011; Barnes et al., 2014), which would affect air quality. In addition, early studies have associated the Arctic sea ice loss with an increasing frequency of atmospheric blocking due to a slow-down flow (Liu et al., 2012). However, the link between the Arctic amplification and weather extremes is complex and no significant trends have been reported (Wollings et al., 2018; Barnes et al., 2012). It must also be acknowledged that the trends of the respective variables, ozone and temperature, were not taken into account. While maximum temperature have shown upwards trends for the past decades (Jacob, 2013), the trends of surface ozone concentrations over Europe are not clear. Previous trend analysis showed a clearer decreasing trend of ozone peaks during the period 2000–2008 over most of the European sites, but no significant trends were found for the recent period 2009–2018 (EEA, 2019). Since our main objective focuses on the dependence between ozone and temperature, we might expect that changes in their relationship could be also reflected in the impacts of atmospheric blocking. However, due to the

complexity in the temperature dependence of ozone (Pusede et al., 2014; Otero et al., 2021) and the changing emissions of ozone precursors, further analysis should be required to investigate the influence of persistent atmospheric conditions while accounting for changes in the temperature-ozone relationship. In spite of this limitation, our results are in a good agreement with previous works that examined the individual effects of blocking on either temperature (Sousa et al., 2018; Pfahl and
Wernli, 2012) or ozone (Ordóñez et al., 2017). Moreover, here we provide a first quantification on the impacts of blocks on compound events of ozone and temperature extremes.

Therefore, an important implication from our findings is the significant influence of atmospheric blocking in the co-occurrence of ozone and temperature extremes in certain European regions. Given the strong linkage between atmospheric blocking and the compounding effect of ozone and temperature extremes, the frequency of blocking events might be used as a key predicting
factor for assessing the health-related risks of the combined effects of ozone pollution and temperature extremes.

*Code and data availability.* Observational ozone data used in this study are available at the Airbase database of the European Environment Agency (EEA) data service: https://www.eea.europa.eu/data-and-maps/data/aqereporting-8.

The ERAInterim reanalysis products are available available on the Climate Data Store (CDS) cloud server: https://cds.climate.copernicus.eu.

The code applied is available on reasonable request from the corresponding author.

*Author contributions.* NO designed the study and performed the statistical analyses with input from H.R. and O.J. NO drafted the manuscript with the contribution of all authors.

*Competing interests.* The authors declare no competing interests.

*Acknowledgements.* We acknowledge Andy Richling for providing the Blocking Index data. This publication was financially supported by Geo.X, the Research Network for Geosciences in Berlin and Potsdam (grant no. SO_087_GeoX). This work was hosted by IASS Potsdam,
with financial support provided by the Federal Ministry of Education and Research of Germany (BMBF) and the Ministry for Science, Research and Culture of the State of Brandenburg (MWFK). The authors would like to sincerely thank the two anonymous reviewers whose comments led to the improvement of this manuscript.

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
