# Peer review of "The impact of atmospheric blocking on the compounding effect of ozone pollution and temperature: A copula-based approach"

_Atmospheric Chemistry and Physics, 2021_

## Referee Comment (RC2)

Review of "The impact of atmospheric blocking on the compounding effect of ozone pollution and temperature: A copula-based approach," by Otero et al.

In this paper, Otero et al. apply a copula-based statistical method to examine the joint dependence of temperature and surface ozone on atmospheric blocking conditions over Europe. The copula approach allowed the authors to model the dependence between two variables, independent of their marginal distributions. Using a 17-year dataset (1999-2015), the authors found that blocks enhance the probability of compound ozone and temperature events by 20% for many stations, especially in central Europe. The presence of an atmospheric block also increases the probability that either an ozone or a temperature exceedance occurs by 30%. Finally, the authors determined that in north-western Europe, given high surface temperatures, blocking increases the probability of ozone exceedances by 30%.

The represents a useful addition to the literature on the influence of meteorology on ozone and surface temperature. I recommend acceptance after key issues are addressed.

**Major comments.**
1. The authors should do more to emphasize how their work builds on that of previous work linking surface ozone over Europe to block episodes. For example, Ordonez et al. (2017) presented a detailed analysis of the impacts of high-latitude blocks on surface ozone in this region. What knowledge gaps occur in the Ordonez et al. (2017) analysis that the current paper begins to fill in?

2. The paper often lacks convincing explanations for the results that emerge from their analysis. Why is surface ozone in some regions so much more sensitive to blocking than in other regions? Do local emissions play a role or typical meteorological conditions? How do the magnitudes of extremes differ across regions and why? What role does geography play – i.e., altitude, latitude, or proximity to oceans?

3. The reader is curious whether trends in ozone, temperature, or blocking episodes have occurred over the 17-year period. Were these trends taken into account when calculating anomalies? How do such trends affect the analysis? Also, were the anomalies calculated with respect to particular days or to the entire season? Was the seasonal cycle somehow accounted for in the different variables?

4. What are the implications of the results? The discussion should consider what impacts anthropogenic climate change will have on the frequency or duration of blocking events. For example, are such events tied to what is known as Arctic Amplification, in which the decrease of the meridional temperature gradient leads to disturbances in the polar jet stream? The issue of Artic Amplification is controversial. For example, see Barnes (2013) and Mann et al. (2018).

Barnes, E .A. (2013), Revisiting the evidence linking Arctic amplification to extreme weather in midlatitudes, *Geophys Res Lett. 40,* (17), 4734–9.

Mann, M. E. (2018), Projected changes in persistent extreme summer weather events: The role of quasi-resonant amplification, *Sci. Adv, 4,* eaat3272.

5. The paper is generally well-written and clear, but lapses in English occur with a frequency of about 4-5 per page.

**Minor comments.**

Abstract. The text should state the time period examined.

Line 29. Over what region did Zhang et al. (2017) perform their analysis?

Lines 66-76. Here the authors should state more clearly how their analysis builds on previous research.

Line 114. References for the different copula functions applied (Clayton, Gumbel and Joe) would be helpful. Another useful reference for the reader unfamiliar with copula functions is Tilloy et al. (2019).

Tilloy, A. (2019), A review of quantification methodologies for multi-hazard interrelationships, *Earth-Science Reviews, 196*, 102881.

Line 117. The reader wonders why it is important to capture lower-tail dependence when the focus of the paper is on extremely high ozone and temperatures.

Line 118. The text should make clear that the best-fit copula function is chosen separately for each measurement site.

Table 1. Symbols used in this table should be defined in a footnote.

Lines 166- 167. The reader is curious if the authors took into account the long-term trends in calculating the anomalies. Also, are the anomalies with respect to particular days or to the entire ozone season? Are the values in Figure 1 the average anomalies at each site calculated over all B1 days in the 17-year time period?

Line 169. Many readers are more familiar with surface ozone reported in ppbv. The authors should consider providing approximate equivalents between $\mu g \ m^{-3}$ and ppbv in a few places.

Figure 1. The caption should indicate that these are average anomalies over the 1999-2015 period. It looks like the anomalies at all sites show statistically significant differences with the mean. Is that right? If yes, is the black contour needed?

Line 181. The text states: "The percentage of Tmax extremes coincident with blocks increases north and eastwards, [which] is consistent with subsidence processes and clear-sky radiative forcing associated with summer blocking events." This is the first of many incidences in which the text fails to really explain why spatial differences in the response to blocking arise.

Figure 2. Numbers in the color bar shows too many digits.

Lines 210-211. Why are certain copula functions exhibit the best fit for some sites, and other functions at other sites? What can we learn from the function selection?

Line 218. Not all readers will know what Benelux refers to.

Lines 233-235. In discussing the strong sensitivity of surface ozone and temperature over the UK to blocking events, the text states "our results also indicate that such combination mainly occurs under blocks likely due to the clear-sky radiative forcing as pointed out by earlier work (Brunner et al., 2017) and subsidence processes associated with the anticyclonic circulation." Again, what is special about the UK compared to other regions?

Supplement. The captions in the supplement are not very explanatory. Units are missing. Figure labels are mostly too tiny to read.

---

## Author Comment (AC1)

**Response to referee comments on acp-2021-297**

The impact of atmospheric blocking on the compounding effect of ozone pollution and temperature: A copula-based approach

Noelia Otero, Oscar Jurado, Tim Butler and Henning W. Rust

**General comment**

We would like to thank both reviewers for their constructive comments and their time spent in reviewing the manuscript. We have carefully revised the manuscript according to their comments. Here, we provide our responses. In each case, we have copied the referees comments (in bold) and our responses are written in standard script. We believe that the changes made in response to the referees comments helped to improve the manuscript. We hope the editor and referees find the revised version suitable for publication in ACP. We also append a marked-up version of the manuscript with the changes mentioned in our responses to the referees. Text deleted is shown as cross out sentences, and extra or new text in red script.

**Response to Referee #1**

**The study looks into the influence of atmospheric blocking on the occurrence of temperature and surface ozone extremes in Europe. It considers both, temperature extremes and ozone extremes alone as well as their co-occurrence using three cases: AND, OR, and Conditional. For all cases the authors show a influence of blocking on the occurrence probabilities. The manuscript addresses the very timely topic of compound events and adds to the research in this field. It is generally well written and the figures are well-made and informative. However, it would also benefit from careful proofreading. I found a number of small mistakes in grammar and content. Also some figures miss units and some could use improvement in several details as by my minor comments. On a technical level I find that the description of methods lacks detail at several points as pointed out below. My only major comment is therefore conditional on my interpretation of temperature extremes: I assume that the authors treat all days equally in their distribution (e.g., in line 177/figure 2 for the exceedances of the 95th percentile of temperature). To me this is flawed as the authors consider the months April-September and the probability for temperature extremes is not equally distributed across these months. So, e.g., days in July are way more likely to be in the upper tail (even if they are not exceptional), while this might even be impossible for days in April. This should to be clarified and/or resolved.**

Thank you very much for this comment. The threshold interpretation (Fig. 2) of the referee is correct. We defined temperature extremes based on the 95th percentile of the distribution during the period of study (1999-2015). We agree with the referee that temperature extremes will be in the upper tail more likely in July or August than in April. However, we would like to highlight that the main objective of this study is to investigate the impacts of blocks on the dependence between temperature and ozone over the whole ozone season. To do that, we have presented a risk analysis based on the joint probability of exceedances (>95th) derived from the copula models, for which a time-(in)dependent threshold does not really affect, as we use the complete data set (April-September 1999-2015). Moreover, it is worth noting that restricting the analysis to summer months would lead to more limitations in terms of the length of the dataset when modelling the cases BI=1 and BI=0. Assessing in more detail seasonal impacts of blocking in the compounding effect of ozone and temperature using a time-varying threshold approach would be definitively an interesting future work.

**Minor comments: line 78: "Maximum daily average 8h [...] concentrations" I am sorry but it is unclear to me what that means exactly.**

Maximum daily average 8h (MDA8O3) is the maximum of daily surface ozone concentration averaged over 8 hours (e.g. rolling average every 8 hours during the day).

**l86: How are the gridded temperatures and station data of ozone brought together? Would it not be better to use ERA5 or even E-Obs as dataset for the temperature? Both are on a 0.25 degree grid and E-Obs is closer to the observations.**

The temperature data was obtained from ERA-Interim (1°x1°) and for each ozone station we extracted the closest grid cell. We agree that ERA5 or E-Obs would provide a higher spatial resolution. However, we do not expect that the use of ERA5 or E-Obs would change the conclusions of the present study.

**l95: I assume it should be 10^6? Similar for l97.**

Yes, thank you. It has been corrected.

**l104: "For two random variables random variables"**

Corrected.

**l113: Could the authors explain in a bit more detail what BI=1 means? Is blocking only considered if it is co-located similar to, e.g., Pfahl et al. 2012 or in a larger region similar to, e.g., Brunner et al. 2017?**

Here, we follow a similar approach than Pfahl and Wernli. (2012) and BI=1 refers to a blocking situation in the same location under consideration for the analysis of the join behavior of temperature and ozone. We have clarified this in the revised version of the manuscript.

**Figure 1: - "calculated with respect the MDAO3 concentrations over the whole period" I assume temperatures are not relative to MDAO3 concentrations? - Do I assume correctly that there are hardly any non-significant cases? I only found one in Ireland. - In Ireland there seem to be two dots missing for ozone but that could be due to the fact that they are white and do not have a contour? - This is just a suggestion but maybe it would be more insightful to provide relative anomalies or anomalies in StdDevs in order to make clear how far outside the norm conditions during blocking are?**

We realise that the caption was not clear, which led to the confusions regarding the anomalies calculation. We apologise for that. The MDA8O3 anomalies were computed with respect to the MDA8O3 concentrations over the whole period of study under blocking situations. Similarly, the temperature anomalies were computed with respect to the temperature over the whole period of study (independently of MDA8O3). The caption and the text related to Fig.1 have been accordingly modified in the revised version. The anomalies were found statistically significant (at the 95% confidence level) in most of the stations: black contour indicates the statistical significance. We thank the referee for the suggestion. We consider that the anomalies are a good indicator in this case and it clearly show the impact of blocks on both MDA8O3 and temperature, consistent with previous studies (e.g Ordóñez et al. (2017)).

**l173: "west-east gradient" That is not obvious to me from figure 1b. Could also be a northsouth gradient, right? If the authors want to make this argument, I would suggest to provide a (densityweighted) average for west vs east (or other regions) to support that argument. Also from briefly looking into Brunner et al. 2017 they also do not seem to support this statement (or maybe the citation is only referring to the radiative heating?)**

Thank you for this comment. We have carefully revised Fig.1 and we agree with the referee that while there is a clear region of high positive anomalies in the central eastern stations, the west-east gradient is not obvious. Therefore, in the revised version of the manuscript we have avoided to make use of a gradient statement that is not completely supported by Fig.1.

**figure1: It should also be noted again that the number of days going into the blocked average shown can differ for each station depending (I assume).**

Thank you for this comment. We have stressed this point in the revised version. As mentioned in the comment above, the caption of Fig.1 has been modified. We hope that the changes help to clarify the questions related to Fig.1.

**Figure S2: Missing units**

Figure S2 has been updated.

**l177: Do I assume correctly that the threshold criterion (95th percentile) is only variable spatially? It should be noted that when using the entire period (rather than say a window around the day of the year considered) for temperature most exceedances will be in summer and not in April, May or September, correct?! This might lead to not very extreme (relatively speaking) summer days being selected over exceptional days from, e.g., April.**

Yes, the threshold criterion vary in space (separately for each station). It is expected that a large number of temperature exceedances occur in summer. However, we would like to clarify that we consider an extended season (i.e. April-September) due to the increasing levels of ozone. Moreover, it must be noted the important role of blocking in high temperatures in spring (e.g. Brunner et al. (2017)). As we aim to model the dependence between temperature and ozone under the influence of blocks, we consider appropriate the use of the whole ozone season and not only the summer months.

**Figure 2: - Is there a reason to use a diverging colorbar in this figure? This seems to suggest that somehow blue dots have less blocking than the climatology at first glance with is not the case, right? - Could the authors indicate significance here as well? (E.g., by using a bootstrap approach?) It seems to me that maybe in the UK there might be some non-significant values for ozone? (around 20% blocked days versus about 15% blocking climatology) - I personally find the non-integer categories of the colorbar confusing, is there a reason for that?**

We believe that the referee might have misunderstood Fig. 2. Fig.2 shows the percentage of exceedances of $MDA8O_3$ (and temperature, Fig.2, right) that are blocked (i.e. $MDA8O_3 > 95th$/total blocked days).For example, in the UK the 20% correspond to the number of days that ozone exceeds the 95th with respect the blocking climatology. We did perform a significance test, and all the values were found to be statistically significant, thus we believe that there is no need to add contours here. The color bar has been updated using different colors. We have clarified in the revised version of the manuscript that the frequency of blocked extremes of ozone and temperature days indicates the percentage of exceendances of MDA8O3 and Tmax with respect the total number of blocked days.

**Figure 3: The category 0-0.1 indicates positive correlation and should be in warm colors I assume?**

Figure 3 has been updated.

**Figures 3 and 4: Since the authors are mostly interested in the difference between the blocked and unblocked cases, would it not be better to just show this differences instead of the two cases beside each other? Or all three options like in figure 5?**

Thank you for this suggestion. We have added the difference plots in Figure 3 and 4 (as shown in Figure 5).

**Figure S5: "Bold black lines" Maybe change to "bold black boxes?"**

It has been modified.

**Figure 5: "Black contours represent statistically significance at the confidence level of 95%" Only for the right column, I assume?**

Yes, that is correct. It has been clarified in the revised version.

**l226: Could the the authors put this in context? I assume that 2.5% is still significantly more than we would expect for to completely uncorrelated variables?**

Both temperature and ozone are by nature two correlated variables. The analysis presented here compare the joint probabilities under the situations of interest, with a without blocks. Then, 2.5% might be significantly

higher than if we compare two uncorrelated variables and we are comparing the probability when there is no blocks (BI=0) (~2.5%) with the values when there is a co-located block (e.g. Fig 5. a).

**l245: Have the authors also investigated the XOR case (assuming this is not what is done in the second cased discussed here): so cases where only and only one of the two is true. I think it would be interesting to look into the spatial pattern of that as well if it is not too much effort.**

Thank you for this comment. No, we did not investigate the XOR case, as our main goal is to examine the exceedances of both ozone and temperature. Here, we have used the main hazard scenarios (Tilloy et al., 2019) to assess the impact of blocks on both ozone and temperature. The XOR case could be considered in a future study.

**l285: Neither of the two studies cited here seem to support/investigate the statement that 70% of temperature extremes coincide with blocking in northern Europe to my knowledge (without having re-read them in detail). Could the authors double-check that?**

We have rephrased the sentence. Our results are consistent with previous studies that showed similar patterns in terms of temperature-blocked extremes over Europe.

**l393: Please fix: "https://doi.org/Online at: https://freva.met.fuberlin.de/about/blocking/"**

This has been corrected.

**Response to Referee #2**

**Review of "The impact of atmospheric blocking on the compounding effect of ozone pollution and temperature: A copula-based approach," by Otero et al. In this paper, Otero et al. apply a copula-based statistical method to examine the joint dependence of temperature and surface ozone on atmospheric blocking conditions over Europe. The copula approach allowed the authors to model the dependence between two variables, independent of their marginal distributions. Using a 17-year dataset (1999-2015), the authors found that blocks enhance the probability of compound ozone and temperature events by 20% for many stations, especially in central Europe. The presence of an atmospheric block also increases the probability that either an ozone or a temperature exceedance occurs by 30%. Finally, the authors determined that in northwestern Europe, given high surface temperatures, blocking increases the probability of ozone exceedances by 30%. The represents a useful addition to the literature on the influence of meteorology on ozone and surface temperature. I recommend acceptance after key issues are addressed.**

**Major comments. 1. The authors should do more to emphasize how their work builds on that of previous work linking surface ozone over Europe to block episodes. For example, Ordonez et al. (2017) presented a detailed analysis of the impacts of high-latitude blocks on surface ozone in this region. What knowledge gaps occur in the Ordonez et al. (2017) analysis that the current paper begins to fill in?**

Thank you for this comment. Yes, Ordóñez et al. (2017) provides a comprehensive analysis of the seasonal impacts of blocks on European surface ozone concentrations. However, our study aims to examine the impacts of block on the co-occurrence of ozone and temperature. To our knowledge this is the first quantification of the impacts of atmospheric blocking on the joint distribution of surface ozone and temperature. We have added some extra text in the revised version of the manuscript to highlight our contribution to the field.

**2. The paper often lacks convincing explanations for the results that emerge from their analysis. Why is surface ozone in some regions so much more sensitive to blocking than in other regions? Do local emissions play a role or typical meteorological conditions? How do the magnitudes of extremes differ across regions and why? What role does geography play – i.e., altitude, latitude, or proximity to oceans?**

As stated in the manuscript, previous studies have shown a regional dependence between ozone and temperature, with a stronger relationship in some European regions (e.g central Europe, see Otero et al. (2016)). Thus, we could anticipate a major influence of blocking on the co-occurrence of ozone and temperature exceedances over those regions. Similarly, blocking has showed to have a regional impact in European heatwaves (e.g. Brunner et al. (2017)). As mentioned in the previous comment, our aim is to quantify the impact of blocks on such dependence between ozone and temperature. We agree that local emissions might play a role for the ozone exceedances in some regions, however how local emissions influence the ozone sensitivity to temperature is not straightforward due to their complex and non-linear relationship (Otero et al. (2021)). Mechanistic explanations of these effects are best produced with numerical models which include representations of the relevant mechanisms. Since the present study employs purely statistical techniques, mechanistic explanations for the effects cannot be produced, but this remains an interesting topic for future work.

**3. The reader is curious whether trends in ozone, temperature, or blocking episodes have occurred over the 17-year period. Were these trends taken into account when calculating anomalies? How do such trends affect the analysis? Also, were the anomalies calculated with respect to particular days or to the entire season? Was the seasonal cycle somehow accounted for in the different variables?**

For both variables, we have calculated the anomalies with respect their corresponding climatology during the ozone season (i.e. April-September) for the whole period of study 1999-2015. As we aimed to model the dependence between ozone and temperature for both situations (BI=1 and BI=0), we decided to use the whole ozone season, with a greater number of observations for both cases (BI=1 and BI=0). Thus, to be consistent with the modelling analysis the anomalies were calculated for the same season.

Unlike the trends for the temperature, previous works pointed out not clear trends for surface ozone concentrations over the past decades in Europe. Ozone peaks have shown a clearer decrease in some European sites during the period of 2000-2008, but not significant trends were reported for the last decade (EEA (2019), Otero et al. (2021)). Similarly, the literature indicates that differences in the frequency of blocks may arise from different blocking detection methods (Wollings et al. (2018)).

We have clarified the calculation of the anomalies and we have added some text in the discussion (also in response to the comment below).

**4. What are the implications of the results? The discussion should consider what impacts anthropogenic climate change will have on the frequency or duration of blocking events. For example, are such events tied to what is known as Arctic Amplification, in which the decrease of the meridional temperature gradient leads to disturbances in the polar jet stream? The issue of Artic Amplification is controversial. For example, see Barnes (2013) and Mann et al. (2018).**

**Barnes, E .A. (2013), Revisiting the evidence linking Arctic amplification to extreme weather in midlatitudes, Geophys Res Lett. 40, (17), 4734–9.**

**Mann, M. E. (2018), Projected changes in persistent extreme summer weather events: The role of quasi-resonant amplification, Sci. Adv, 4, eaat3272.**

Thank you for this comment. One of the main implications of our study is the significant role playing by persistent atmospheric blocking on the co-occurrence of ozone and temperature exceedances in some European regions. Besides the strong link shown between blocking and temperature extremes, our results have further indicated that atmospheric blocking significantly enhances the ozone-temperature risks in some European regions. We believe that our findings are relevant, in particular for health risk assessment.

We have taken note of this comment and the references and we have added some extra text in the discussion.

**5. The paper is generally well-written and clear, but lapses in English occur with a frequency of about 4-5 per page.**

Thank you, we have carefully revised the text.

**Minor comments.**

**Abstract. The text should state the time period examined.**

We have added this information now.

**Line 29. Over what region did Zhang et al. (2017) perform their analysis?**

Zhang et al. (2017) mainly focused on the US. Firstly, they assessed to ability of WRF-Chem to reproduce the observed extreme weather events and ozone concentration in the US. Based on this analysis, they extended the discussion in the broader context of the multi-model CMIP5 ensemble. The regions has been now specified.

**Lines 66-76. Here the authors should state more clearly how their analysis builds on previous research.**

We have emphasized our contribution to the field building upon previous analysis.

**Line 114. References for the different copula functions applied (Clayton, Gumbel and Joe) would be helpful. Another useful reference for the reader unfamiliar with copula functions is Tilloy et al. (2019).**

**Tilloy, A. (2019), A review of quantification methodologies for multi-hazard interrelationships, Earth-Science Reviews, 196, 102881.**

Thank you. The reference has been added.

**Line 117. The reader wonders why it is important to capture lower-tail dependence when the focus of the paper is on extremely high ozone and temperatures.**

We have included different families of copulas in order to better capture the different behavior of the variables of interest: ozone and temperature. It is important to include different copula families, as the dependence structure may vary across the stations. In this case, we included the Clayton copulas that can capture lower-tail dependence.

**Line 118. The text should make clear that the best-fit copula function is chosen separately for each measurement site.**

Thank you. We have clarified this in the revised version of the manuscript.

**Table 1. Symbols used in this table should be defined in a footnote.**

The caption of table 1 has been modified.

**Lines 166- 167. The reader is curious if the authors took into account the long-term trends in calculating the anomalies. Also, are the anomalies with respect to particular days or to the entire ozone season? Are the values in Figure 1 the average anomalies at each site calculated over all B1 days in the 17-year time period?**

Please see our previous comment related to the anomalies calculation. The caption of Fig.1 has been accordingly modified.

**Line 169. Many readers are more familiar with surface ozone reported in ppbv. The authors should consider providing approximate equivalents between μg m-3 and ppbv in a few places.**

Thank you for the suggestion. Here, we consider that the use of $\mu g/m^3$ should be clear for the readers, as it commonly used in the literature and it is also the unit required for regulatory reporting in Europe.

**Figure 1. The caption should indicate that these are average anomalies over the 1999-2015 period. It looks like the anomalies at all sites show statistically significant differences with the mean. Is that right? If yes, is the black contour needed?**

The caption of Fig.1 has been modified. Yes, the anomalies were statistically significant at most of the sites. Only a few places did not show statistically significant anomalies. We decided to keep the contour to show those locations.

**Line 181. The text states: "The percentage of Tmax extremes coincident with blocks increases north and eastwards, [which] is consistent with subsidence processes and clear-sky radiative forcing associated with summer blocking events." This is the first of many incidences in which the text fails to really explain why spatial differences in the response to blocking arise.**

As stated at the beginning of section 3, atmospheric blocking shows a clear seasonal variability: maximum occurrences in late winter and early spring and minimum frequencies in late summer and early autumn (Barriopedro et al., 2006). Moreover, it has been shown an enhanced of atmospheric blocking over eastern Europe in summer (Barriopedro et al., 2006), which is also reflecting by our results, showing a large number of temperature exceedances over the eastern locations. We have added some extra text in the revised version of the manuscript.

**Figure 2. Numbers in the color bar shows too many digits**

Figure 2 has been updated.

**Lines 210-211. Why are certain copula functions exhibit the best fit for some sites, and other functions at other sites? What can we learn from the function selection?**

The relationship between ozone and temperature is by nature complex (Pusede et al. (2014)). Therefore, one could expect differences in their dependence structure across sites capture by different copula families. The selection procedure is merely based on a data-driven approach. As mentioned in section 2, the copula families included in this study are commonly used as they can capture a variety of joint dependence, such as asymmetrical tail dependence (e.g. Clayton or Gumbel) or symmetrical tail dependence (e.g. t-copula). From the copula selection it can be observed the different dependence structure in the selected locations.

**Line 218. Not all readers will know what Benelux refers to.**

The region of Benelux has been specified in the revised version.

**Lines 233-235. In discussing the strong sensitivity of surface ozone and temperature over the UK to blocking events, the text states "our results also indicate that such combination mainly occurs under blocks likely due to the clear-sky radiative forcing as pointed out by earlier work (Brunner et al., 2017) and subsidence processes associated with the anticyclonic circulation." Again, what is special about the UK compared to other regions?**

As shown in the manuscript, our analysis shows that blocking notable increase the dependence between ozone and temperature. While we could expect an impact in the central regions based on previous work (Otero et al. (2016)), the copula analysis revealed that the atmospheric blocking greatly influence the co-occurrence of temperature and ozone extremes. This is already emphasize in the manuscript.

**Supplement. The captions in the supplement are not very explanatory. Units are missing. Figure labels are mostly too tiny to read.**

Thank you. We have updated the caption and some of the figures of the Supplement.

**References**

Barriopedro, D., García-Herrera, R., Lupo, A.R., Herández, E.R.R., 2006. A climatology of northern hemisphere blocking. J Clim 19, 1042–1063. https://doi.org/10.1175/JCLI3678.1

Brunner, L., Hegerl, G.C., Steiner, A.K., 2017. Connecting atmospheric blocking to european temperature extremes in springs. J Clim. 30(2), 585–94. https://doi.org/10.1175/JCLI-D-16-0518.1

EEA, E.E.A., 2019. Air quality in europe-2019 report. Tech. rep.

Ordóñez, C., Barriopedro, D., García-Herrera, R., Sousa, P.S., Schnell., J.L., 2017. Regional responses of surface ozone in europe to the location of high-latitude blocks and subtropical ridges. Atmos. Chem. Phys. 17, 3111–3131. https://doi.org/10.5194/acp-17-3111-2017

Otero, N., Rust, H.W., Butler., T., 2021. Temperature dependence of tropospheric ozone under NOx reductions over germany,. Atmospheric Environment 253, 1352–2310. https://doi.org/https://doi.org/10.1016/j.atmosenv.2021.118334.

Otero, N., Sillmann, J., Schnell, J.L., Rust, H.W., Butler., T., 2016. Synoptic and meteorological drivers of extreme ozone concentrations over europe. Environ. Res. Lett. 11, 024005. https://doi.org/10.1088/1748-9326/11/2/024005

Pfahl, S., Wernli., H., 2012. Quantifying the relevance of atmospheric blocking for co-located temperature extremes in the northern hemisphere on (sub-)daily time scales. Geophysical Research Letters 39, L12807. https://doi.org/10.1029/2012GL052261

Pusede, S.E., Gentner, D.R., Wooldridge, P.J., Browne, E.C., Rollins, A.W., Min, K.-E., Russell, A.R., Thomas, J., Zhang, L., Brune, W.H., Henry, S.B., DiGangi, J.P., Keutsch, F.N., Harrold, S.A., Thornton, J.A., Beaver, M.R., Clair, J.M.St., Wennberg, P.O., Sanders, J., Ren, X., VandenBoer, T.C., Markovic, M.Z., Guha, A., Weber, R., Coldstein, A.H., Cohen, R.C., 2014. On the temperature dependence of organic reactivity, nitrogen oxides, ozone production, and the impact of emission controls in san joaquin valley, california. Atmos. Chem. Phys. 14, 3373–3395. https://doi.org/10.5194/acp-14-3373-2014

Tilloy, A., Malamud, B.D., Winter, H., Joly-Laugel, A., 2019. A review of quantification methodologies for multi-hazard interrelationships. Earth-Science Reviews 196, 0012–8252. https://doi.org/https://doi.org/10.1016/j.earscirev.2019.102881

Wollings, T., Barriopedro, D., Methven, D., Son, S.W., Martius, O., Harvey, B., Sillmann, J., Lupo, A.R., Seneviratne, S., 2018. Blocking and its response to climate change. Curr Clim Change Rep 4, 287–300. https://doi.org/10.1007/s40641-018-0108-z

Zhang, H., Wand, Y., Park, T.W., Deng, Y., 2017. Quantifying the relationship between extreme air pollution events and extreme weather events. Atmos. Res. 188, 64–79. https://doi.org/10.1016/j.atmosres.2016.11.010

---

## Referee Report (RR1)

In this study, Otero et al. examine the impact of atmospheric blocking on the ozone and temperature as measured at 300 stations across Europe during the period 1999-2015. The authors apply a copula-based method to model the probabilities of extreme temperature and ozone under blocking and non-blocking conditions. The approach allows the authors to examine the impact of blocks on the joint probability distribution of these two variables. The results show that blocking increases the probability of co-occurrent ozone and temperature exceedances by 15%-20% at the stations in central, north-west and east of Europe. The probability of combined ozone and temperature extremes under non-blocking conditions is small everywhere.

The authors did not adequately address many of the criticisms from the reviewers. In some cases, the authors addressed the reviewers' comments only in the response document, and not in the text itself. This means that only the reviewers will see the authors' reasoning, and not the general readership. The paper should acknowledge the limitations pointed out by the reviewers. I recommend major revisions.

**Main criticisms.**

1. Reviewer 1 stated: "I assume that the authors treat all days equally in their distribution (e.g., in line 177/figure 2 for the exceedances of the 95th percentile of temperature). To me this is flawed as the authors consider the months April-September and the probability for temperature extremes is not equally distributed across these months. So, e.g., days in July are way more likely to be in the upper tail (even if they are not exceptional), while this might even be impossible for days in April. This should to be clarified and/or resolved."

The reviewer has raised an important point. The authors address this point in the response, saying, "We agree with the referee that temperature extremes will be in the upper tail more likely in July or August than in April." But the authors have not revised the text sufficiently. It's true a new sentence has been added: "For both variables, we have calculated the anomalies with respect [to] their corresponding climatology during the ozone season (i.e. April-September) for the whole period of study 1999-2015." But does this mean that daily anomalies were calculated – e.g., the anomaly on April 1, 2015, at a particular station relative to all April 1 data at that station? In fact, both reviewers were confused about the construction of the anomalies.

What I think the authors mean is that the anomalous temperature and ozone values were calculated separately for each ozone season, with the 95th percentile of each variable comprising the highest ~10 daily temperatures and ozone concentrations during the 183 days between April 1 and September 30. Is that right?

Importantly, the authors also need to acknowledge in the text the weakness of their approach to construct anomalies, as pointed out by Reviewer 1. They should provide (in the text) their rationale for applying this approach despite its weakness.

2. Reviewer 2 asked for greater clarification on how this paper builds on previous work. The revised text does indeed begin to address this request. Can the authors also say that the copula-based approach allows a quantification of the probability of joint exceedances?

3. Reviewer 2 also asked for some discussion of the drivers of the spatial patterns of the relationships between blocking conditions, ozone, and temperature. The authors responded that this was beyond the scope of the text, saying "Since the present study employs purely statistical techniques, mechanistic explanations for the effects cannot be produced, but this remains an interesting topic for future work." I disagree. There exist many papers relying on statistics that also offer mechanistic explanations for results. The literature is there – e.g., Ordonez et al. (2017) and Sousa et al. (2018). What accounts for the spatial variation in the drivers of high ozone and temperature? Are there other drivers besides blocks for extreme ozone or temperature events? There is some discussion of subtropical ridges but it is hard to follow. The reader is curious and seeks more than just a reporting of results.

An example of text that provides insufficient interpretation is the following (Lines 318+): "However, we found a significant increase in the conditional probability over the north-west stations and a slight increase over the central-east stations [Figure 5f]. This suggests that over such regions ozone extremes tend to occur given high temperatures which are strongly connected with atmospheric blocking. This is likely due to the position of the block during the ozone season covering spring and summertime when the increased solar radiation lead to warm temperature in the blocked regions." Where exactly is the block position? Wouldn't the relationship of high temperatures leading to high ozone hold true throughout the domain? Figure S1 shows the frequency of blocks during the ozone season, but that is not the same as "the position of the block." In any event, the pattern in Figure S1 is not similar to that in Figure 5f. Finally, does the duration of the blocks have an impact on the copula results?

I recommend that Section 3.1 (Impact of atmospheric blocking on ozone and temperature) and Section 3.2 (Copula results) each begin with a detailed description of the results and then conclude with a short paragraph interpreting the results for that section. The interpretation would include an account of spatial variability of all results. There exists sufficient literature for the authors to begin to interpret this spatial variability, though there will also likely be gaps in our knowledge. Citations to other papers should briefly describe the mechanisms that these papers suggest.

4. Reviewer 2 asked about the impact of trends on the results. In response, the authors again state that "For both variables, we have calculated the anomalies with respect [to] their corresponding climatology during the ozone season (i.e. April-September) for the whole period of study 1999-2015." As stated in #1 above, I think that means that the anomalies are calculated with respect to all values recorded during each ozone season separately (and not in fact over the "whole period of study").

In any event, if trends in either ozone or temperature have occurred, then the extremes may become more (or less) extreme over the 17 years of study, and that could muddy the relationship of blocking conditions and these variables. For example, Yan et al. (2019) find a rapid decline of relatively high ozone concentrations from 1995-2012, especially in rural areas. At the very least, the authors need to acknowledge these trends in the text and consider the impact of these trends on their analysis.

Yan, Y., J. Lin, A. Pozzerc, S. Konga, and J. Lelieveld (2019), Trend reversal from high-to-low and from rural-to-urban ozone concentrations over Europe, Atmos. Env., 213, 25-36.

5. Reviewer 2 commented that lapses in English occur with a frequency of 4-5 per page. These lapses are still there – e.g., "especifically," "the probabilities associated to…," improper use of "with respect" and "allow to," "Artic," "bock," and many others. The authors should employ an editor to fix these minor but distracting errors.

6. Reviewer 2 commented that Figure labels are too tiny to read. They continue to be too tiny, both in the main text and the Supplement. For example, the tiny B=0 and B=1 text at the top of many panels is so small that it's easy to miss. Numbers and units beside the color bars are also tiny.

**Minor comments.**

Line 104. "The BI was computed through the Free Evaluation System Framework (see Richling et al. (2015) for more details), specifically with the single plug-in corresponding to the blocking-2d (Freva, 2017)." The average reader will not understand this sentence.

Table 1. There appear to be some typos in this table – e.g., in the Clayton equation.

Equations 2-4. The variables u and v should be defined here, not just in the Table.

Line 271-2. The text states: "For the COND probability, both the computation domain and the critical region evolve when moving along higher temperatures…" This is not clear. What is meant by "critical region" and "computation domain"?

---

## Author Response (AR2)

**Response to referee comments on acp-2021-297**

**The impact of atmospheric blocking on the compounding effect of ozone pollution and temperature: A copula-based approach**

Noelia Otero, Oscar E. Jurado, Tim Butler and Henning W. Rust

**General comment**

We would like to thank the editor and referees for their comments on the manuscript.We really appreciate the time spent in reviewing the manuscript and we apologise that we did not adequately address the comments before. In this second review stage, we have carefully revised the main criticisms and in response to this, substantial changes have been done in the manuscript. Please note that the changes have been applied in the revised version. A marked-up version of the manuscript is included. Changes are shown in red script and the deleted text is shown as cross out sentences. Here we provide our responses. The comments from the referees are in bold script and our responses are given in standard script.

**Response to Referee #2**

**Main criticisms.**

**1. Reviewer 1 stated: "I assume that the authors treat all days equally in their distribution (e.g., in line 177/figure 2 for the exceedances of the 95th percentile of temperature). To me this is flawed as the authors consider the months April-September and the probability for temperature extremes is not equally distributed across these months. So, e.g., days in July are way more likely to be in the upper tail (even if they are not exceptional), while this might even be impossible for days in April. This should to be clarified and/or resolved."**

**The reviewer has raised an important point. The authors address this point in the response, saying, "We agree with the referee that temperature extremes will be in the upper tail more likely in July or August than in April." But the authors have not revised the text sufficiently. It's true a new sentence has been added: "For both variables, we have calculated the anomalies with respect [to] their corresponding climatology during the ozone season (i.e. April-September) for the whole period of study 1999-2015." But does this mean that daily anomalies were calculated – e.g., the anomaly on April 1, 2015, at a particular station relative to all April 1 data at that station? In fact, both reviewers were confused about the construction of the anomalies.**

**What I think the authors mean is that the anomalous temperature and ozone values were calculated separately for each ozone season, with the 95th percentile of each variable comprising the highest ~10 daily temperatures and ozone concentrations during the 183 days between April 1 and September 30. Is that right?**

**Importantly, the authors also need to acknowledge in the text the weakness of their approach to construct anomalies, as pointed out by Reviewer 1. They should provide (in the text) their rationale for applying this approach despite its weakness.**

We realised that there was a misunderstanding about the construction of the anomalies and it was not sufficiently clarified in the previous version. Below, we provide a better explanation about the anomalies and the use of the 95th percentile for defining the extremes. We hope that the following explanations clarify the

referees concerns. We have accordingly modified the text in the revised version of the manuscript, specifically, in Section 3.1 in the following lines (in the marked-up version):

L196-201

We start by examining the impacts of blocks on the anomalies of ozone and temperature (separately), in order to establish a comparison of anomalies across different stations. MDA8O$_3$ anomalies were calculated as the difference between MDA8O$_3$ values and the average of MDA8O$_3$ over all days in April-September in the period of study 1999-2015. This average is obtained individually for every station. Similarly, Tmax anomalies were calculated with respect to the average value over all Tmax values from April-September during the same period. It is important to highlight that all calculations were applied separately for each station, and therefore the number of blocking days might differ across the different stations.

L213-221

To this end, we do not work with anomalies but we fix a threshold for MDA8O$_3$ as well as for Tmax. The absolute values for these thresholds vary among stations. A transparent way to set these thresholds are quantiles, i.e. values with a specified non-exceedance probability. For each station, we use the 0.95-quantile (or 95th percentile) from the sample restricted to April to September and thus get individual thresholds for all stations reflecting their local climatology. In the following, we define days with MDA8O$_3$ exceeding this threshold as extremes. Then, we obtain relative frequencies by dividing the number of extreme days with a simultaneous blocking by the number of total days in the data set restricted to April to September (i.e. $\hat{p} = extremeswithblocking/3111$) .A similar approach was applied in Ordóñez et al. (2017) that calculated the percentage of blocking days with MDA8O$_3$ values above the 90th percentile. It must be noticed that the spatial variability of high levels of ozone is very heterogeneous (Fig. S2).

L224-237

The same procedure using the 95th percentile was applied for identifying days of Tmax exceedances and days above the 95th percentile of the Tmax (Fig. S2) were classified as exceedances. We acknowledge that in the case of Tmax, the number of exceedances above the 95th percentile might be not equally distributed across the ozone season (i.e. this threshold is more likely to be exceeded in July and August than it is in April and September). While this could be corrected by either using a threshold that varies seasonally or by removing the seasonal trend in the data, we would like to stress that the main goal of this study is to quantify the impacts of blocking on the upper-tail dependence between MDA8O$_3$ and Tmax over the entire ozone season. Our main interest is in the physiological effects of such compound events, for which only absolutely high temperatures (as they tend to occur in July or August) are relevant.
As in other studies (Schnell and Prather, 2017), we use the 95th percentile over the period April to September. A lower threshold, e.g. the 90th percentile would lead to many temperature values not being physiological relevant; a higher threshold, e.g. the 99th percentile would lead to a strong reduction in the number of data available for the subsequent copula modelling. The 95th percentile-based definition to examine the individual impacts of blocks on MDA8O$_3$ and Tmax it is also justified to be consistent with the joint probability analysis, for which the 95th percentile is applied for the risk assessment (see below). Moreover, earlier studies used a similar threshold-percentile based definition to assess the links between temperature extremes and atmospheric blocking Pfahl and Wernli. (2012).

**2. Reviewer 2 asked for greater clarification on how this paper builds on previous work. The revised text does indeed begin to address this request. Can the authors also say that the copula based approach allows a quantification of the probability of joint exceedances?**

Thank you for this comment. As stated in the introduction, previous works examined the individual impacts of atmospheric blocking on surface ozone (Ordóñez et al., 2017) or temperature (Pfahl and Wernli., 2012), but our study provides for the first time (to our knowledge) a quantification of the impacts of blocks in the compounding effect of ozone and temperature. Indeed, copulas enable to assess their joint dependence. Therefore, we are confident that the copula-modeling approach is robust and solid to quantify the effects of atmospheric blocking. Extra text has been added in the revised version (i.e. in the introduction section, L81-84 in the marked-up version).

**3. Reviewer 2 also asked for some discussion of the drivers of the spatial patterns of the relationships between blocking conditions, ozone, and temperature. The authors responded that this was beyond the scope of the text, saying "Since the present study employs purely statistical techniques, mechanistic explanations for the effects cannot be produced, but this remains an interesting topic for future work." I disagree. There exist many papers relying on statistics that also offer mechanistic explanations for results. The literature is there – e.g., Ordonez et al. (2017) and Sousa et al. (2018). What accounts for the spatial variation in the drivers of high ozone and temperature? Are there other drivers besides blocks for extreme ozone or temperature events? There is some discussion of subtropical ridges but it is hard to follow. The reader is curious and seeks more than just a reporting of results. An example of text that provides insufficient interpretation is the following (Lines 318+): "However, we found a significant increase in the conditional probability over the north-west stations and a slight increase over the central-east stations [Figure 5f]. This suggests that over such regions ozone extremes tend to occur given high temperatures which are strongly connected with atmospheric blocking. This is likely due to the position of the block during the ozone season covering spring and summertime when the increased solar radiation lead to warm temperature in the blocked regions." Where exactly is the block position? Wouldn't the relationship of high temperatures leading to high ozone hold true throughout the domain? Figure S1 shows the frequency of blocks during the ozone season, but that is not the same as "the position of the block." In any event, the pattern in Figure S1 is not similar to that in Figure 5f. Finally, does the duration of the blocks have an impact on the copula results?**

**I recommend that Section 3.1 (Impact of atmospheric blocking on ozone and temperature) and Section 3.2 (Copula results) each begin with a detailed description of the results and then conclude with a short paragraph interpreting the results for that section. The interpretation would include an account of spatial variability of all results. There exists sufficient literature for the authors to begin to interpret this spatial variability, though there will also likely be gaps in our knowledge. Citations to other papers should briefly describe the mechanisms that these papers suggest.**

Thank you for this comment. Following the referee suggestion, Sections 3.1 and 3.2 have been modified. In particular, substantial changes have been applied in Section 3.1 to better clarify the construction of the anomalies as well as the extremes identification. Moreover, we conclude each subsection with a short paragraph to summarizes the main results while linking our findings to previous works.

**4. Reviewer 2 asked about the impact of trends on the results. In response, the authors again state that "For both variables, we have calculated the anomalies with respect [to] their corresponding climatology during the ozone season (i.e. April-September) for the whole period of study 1999- 2015." As stated in #1 above, I think that means that the anomalies are calculated with respect to all values recorded during each ozone season separately (and not in fact over the "whole period of study").**

**In any event, if trends in either ozone or temperature have occurred, then the extremes may become more (or less) extreme over the 17 years of study, and that could muddy the relationship of blocking conditions and these variables. For example, Yan et al. (2019) find a**

rapid decline of relatively high ozone concentrations from 1995-2012, especially in rural areas. At the very least, the authors need to acknowledge these trends in the text and consider the impact of these trends on their analysis. Yan, Y., J. Lin, A. Pozzerc, S. Konga, and J. Lelieveld (2019), Trend reversal from high-to-low and from rural-to-urban ozone concentrations over Europe, Atmos. Env., 213, 25-36.

As mentioned in the previous comment, in the revised version we have better clarified the construction of the anomalies. Following the referee suggestion regarding the trends, we acknowledge this limitation in the last section of the manuscript. Specifically in L436-447 (in the marked-up version):

While maximum temperature have shown upwards trends for the past decades (Jacob, 2013), the trends of surface ozone concentrations over Europe are no clear. Previous trends analysis showed a clearer decreasing trend of ozone peaks during the period 2000–2008 over most of the European sites, but not significant trends were found for the recent period 2009–2018 (EEA, 2019). Since our main objective focuses on the dependence between ozone and temperature, we might expect that changes in their relationship could be also reflected in the impacts of atmospheric blocking. However, due to the complexity in the temperature dependence of ozone Otero et al. (2021) and the changing emissions of ozone precursors, further analysis should be required to investigate the influence of persistent atmospheric conditions while accounting for changes in the temperature-ozone relationship. In spite of this limitation, our results are in a good agreement with previous works that examined the individual effects of blocking on either temperature Pfahl and Wernli. (2012) or ozone (Ordóñez et al., 2017).Moreover, here we provide a first quantification on the impacts of blocks on compound events of ozone and temperature extremes.

**5. Reviewer 2 commented that lapses in English occur with a frequency of 4-5 per page. These lapses are still there – e.g., "especifically," "the probabilities associated to. . .," improper use of "with respect" and "allow to," "Artic," "bock," and many others. The authors should employ an editor to fix these minor but distracting errors.**

We apologise for this. The manuscript has been carefully reviewed.

**6. Reviewer 2 commented that Figure labels are too tiny to read. They continue to be too tiny, both in the main text and the Supplement. For example, the tiny B=0 and B=1 text at the top of many panels is so small that it's easy to miss. Numbers and units beside the color bars are also tiny. Minor comments.**

Figures 2,3,4 and 6 in the main text and Figures S1,S2 in the supplementary material have been updated.

**Line 104. "The BI was computed through the Free Evaluation System Framework (see Richling et al. (2015) for more details), specifically with the single plug-in corresponding to the blocking-2d (Freva, 2017)." The average reader will not understand this sentence.**

This sentence has been changed.

**Table 1. There appear to be some typos in this table – e.g., in the Clayton equation. Equations 2-4. The variables u and v should be defined here, not just in the Table.**

Thank you. It has been corrected.

**Line 271-2. The text states: "For the COND probability, both the computation domain and the critical region evolve when moving along higher temperatures. . . " This is not clear. What is meant by "critical region" and "computation domain"?**

Thank you for the comment. We have clarified this sentence with the following extra text: "both the computation domain (i.e. the joint space where the probability of exceedances is calculated ) and the critical region (i.e. the regions of exceedances of ozone conditioned by temperature)". Figure S5 shows the space where the joint probabilities are estimated for the $COND$ case. In addition to the text, we have considered to move Fig.S5 from the supplementary material to the main text (now, Fig. 5) for a graphical explanation of the three hazard scenarios in the manuscript.

**References**

EEA, E.E.A., 2019. Air quality in europe-2019 report. Tech. rep.

Jacob, D., 2013. EURO-CORDEX: New high-resolution climate change projections for european impact research. Reg. Environ. Change 1–16.

Ordóñez, C., Barriopedro, D., García-Herrera, R., Sousa, P.S., Schnell., J.L., 2017. Regional responses of surface ozone in europe to the location of high-latitude blocks and subtropical ridges. Atmos. Chem. Phys. 17, 3111–3131. https://doi.org/10.5194/acp-17-3111-2017

Otero, N., Rust, H.W., Butler., T., 2021. Temperature dependence of tropospheric ozone under NOx reductions over germany,. Atmospheric Environment 253, 1352–2310. https://doi.org/https://doi.org/10.1016/j.atmosenv.2021.118334.

Pfahl, S., Wernli., H., 2012. Quantifying the relevance of atmospheric blocking for co-located temperature extremes in the northern hemisphere on (sub-)daily time scales. Geophysical Research Letters 39, L12807. https://doi.org/10.1029/2012GL052261

Pusede, S.E., Gentner, D.R., Wooldridge, P.J., Browne, E.C., Rollins, A.W., Min, K.-E., Russell, A.R., Thomas, J., Zhang, L., Brune, W.H., Henry, S.B., DiGangi, J.P., Keutsch, F.N., Harrold, S.A., Thornton, J.A., Beaver, M.R., Clair, J.M.St., Wennberg, P.O., Sanders, J., Ren, X., VandenBoer, T.C., Markovic, M.Z., Guha, A., Weber, R., Coldstein, A.H., Cohen, R.C., 2014. On the temperature dependence of organic reactivity, nitrogen oxides, ozone production, and the impact of emission controls in san joaquin valley, california. Atmos. Chem. Phys. 14, 3373–3395. https://doi.org/10.5194/acp-14-3373-2014

Schnell, J.L., Prather, M.J., 2017. Co-occurrence of extremes in surface ozone, particulate matter, and temperature over eastern north america. P. Natl. Acad. Sci. USA 114, 2854–2859. https://doi.org/10.1073/pnas.1614453114

Sousa, P.M., Barriopedro, J.D., Soares, P.M., Santos, J.A., 2018. European temperature responses to blocking and ridge regional patterns. Clim. Dynam. 50(1-2), 457–77. https://doi.org/10.1007/s00382-017-3620-2

---

## Author Response (AR3)

**Response to referee comments on acp-2021-297**

The impact of atmospheric blocking on the compounding effect of ozone pollution and temperature: A copula-based approach

Noelia Otero, Oscar E. Jurado, Tim Butler and Henning W. Rust

**General comment**

We would like to thank the editor and referee for their comments on the manuscript.We appreciate the time spent in reviewing the manuscript. Please note that the changes have been applied in the revised version. A marked-up version of the manuscript is included. Changes are shown in red script and the deleted text is shown as cross out sentences. Here we provide our responses. The comments from the referee are in bold script and our responses are given in standard script.

**Response to Referee #2**

**In this paper, Otero and coauthors examine the impact of atmospheric blocking on the ozone and temperature as measured at 300 stations across Europe during the period 1999-2015. The authors apply a copula-based method to model the probabilities of extreme temperature and ozone under blocking and non-blocking conditions.**

**The authors have now made a concerted effort to address the reviewers' concerns. For example, they now clarify their approach for calculating ozone and temperature anomalies at each measurement station, and they justify that approach. They also discuss the implications of their results given current knowledge of future trends in climate and surface ozone. Finally, they now work harder to interpret the spatial distributions of the probabilities of ozone exceedances under blocking conditions.**

**I recommend publishing once the minor issues below have been addressed.**

**Minor comments.**

**1. The paper has a large number of typos and lapses in English, about 1-2 per page. I recommend that the authors ask for assistance in editing.**

Following your suggestion, we have asked for assistance in editing the manuscript. According to this, some corrections and edits have been applied in the revised version.

**2. The last paragraph of the introduction should state that the paper focuses on Europe.**

We have stressed in the last paragraph of the introduction that our study aims to asses persistent blocks on the compounding effect of ozone and temperature over Europe (L67 in the marked-up version).